# Single-cell sequencing reveals increased LAMB3-positive basal keratinocytes and ZNF90-positive fibroblasts in autologous cultured epithelium

Weiling Lian[1,3], Xuanhao Zeng[1,3], Jian Li[1,3], Qing Zang[1,3], Yating Liu[1], Haozhen Lv[1], Shujun Chen[1], Shiyi Huang[1], Jiayi Shen[1], Luyan Tang[1], Yu Xu[1], Fuyue Wu[2], Qi Zhang [1✉] & Jinhua Xu [1✉]

Autologous cultured epithelium grafting (ACEG) presents a promising treatment for refractory vitiligo, yet concerns regarding infections and immunological reactions hinder its surgical use due to serum and feeder dependencies. Addressing this, we culture autologous epithelium under serum- and feeder-free (SFF) conditions, comparing its safety and efficacy with serum- and feeder-dependent (SFD) conditions in stable vitiligo patients, and we discover no significant differences in repigmentation between the SFF and SFD grafts. Single-cell RNA transcriptomics on SFF- and SFD-cultured epithelium alongside healthy skin reveal increased populations of LAMB3+ basal keratinocytes and ZNF90+ fibroblasts in the SFF sheets. Functional analyses showcase active cellular metabolism in LAMB3+ basal keratinocytes, vital in extracellular matrix homeostasis, while ZNF90+ fibroblasts demonstrate increased differentiation, essential in collagen formation for cell adhesion. Importantly, these cell populations in SFF sheets exhibit enhanced interactions with melanocytes compared to SFD sheets. Further, knockdown experiments of LAMB3 in keratinocytes and ZNF90 in fibroblasts lead to a downregulation in melanocyte ligand-receptor-related genes. Overall, SFF sheets demonstrate comparable efficacy to SFD sheets, offering superior safety. LAMB3+ basal keratinocytes and ZNF90+ fibroblasts act as potential drivers behind repigmentation in ACEG under SFF conditions. This study provides translational insights into ACEG repigmentation and potential therapeutic targets for vitiligo.

[1] Department of Dermatology, Huashan Hospital, Fudan University, Shanghai Institute of Dermatology, Shanghai, China. [2] ReMed Regenerative Medicine Clinical Application Institute, Shanghai, China. [3] These authors contributed equally: Weiling Lian, Xuanhao Zeng, Jian Li, Qing Zang. ✉email: zhangqi@huashan.org.cn; jinhuaxu@fudan.edu.cn

Vitiligo is an acquired hypopigmentation disorder that is characterized by generalized or localized white patches resulting from a decrease in melanocytes. Surgical interventions, including autologous tissue and cellular grafting, can be considered for patients with stable vitiligo who have not responded to drug therapy[1].

In 1975, Rheinwald and Green pioneered the use of irradiated 3T3 fibroblasts to grow single keratinizing cells into stratified squamous epithelium and modified it to be suitable for grafting[2,3]. In this manner, O'Connor et al. extensively grafted cultured epithelium generated from autologous epidermal cells onto patients with burn wounds[4], demonstrating that autologous melanocytes can be grafted to restore extensive pigmentation in hypopigmentation diseases. Kumagai and Uchikoshi[5] reported that repigmentation was successfully achieved in six cases of hypopigmentaion using cultured epithelial autografts. Research over the past two decades has proven the efficacy of this technique in patients with stable vitiligo[6,7]. However, this culture contains mouse-irradiated fibroblast feeders in a medium with fetal bovine serum, which may be contaminated with animal pathogens or feeder cells and be hazardous for clinical application. It has been found that human keratinocytes can propagate without feeder layer cells or serum and reconstitute the epidermis to enhance wound healing[8–10].

To address this problem, we developed a modified autologous cultured epithelium grafting (ACEG) system under serum- and feeder-free (SFF) conditions. Our preliminary studies have verified the safety and efficacy of ACEG under SFF conditions for the treatment of stable vitiligo and identified a comparable repigmentation rate when compared to that of autografts cultured under serum- and feeder-dependent (SFD) conditions, thus providing surgical procedures in over 1000 patients since 2015, with a total efficacy rate of 82.81%[11]. However, the major determinants of the repigmentation mechanism are still unknown.

Single-cell RNA sequencing, as an emerging technology, may provide an opportunity to uncover the cell heterogeneity of epithelial sheets. This method has revealed cell heterogeneity in vitiligo lesions, representing cell populations that promote disease progression, determine autoimmune disorders, and indicate therapeutic targets[12–14]. To identify potential molecular benefits for repigmentation, we conducted single-cell transcriptomics of epithelial grafts under SFF and SFD conditions to assess the similarities and variations in cell composition.

## Results

### The similarities in morphology and efficacy of cultured epithelial sheets under SFF and SFD conditions.
The scheme of the study is depicted in Fig. 1. From November 2020 to January 2021, we approached 17 eligible patients for inclusion in our study cohort, and only three individuals consented to contribute their samples for the investigation (Table 1). Based on previous literature and our own experience with this treatment, factors that can affect the outcome include the age of the patient, the ratio of keratinocytes and melanocytes in the skin sheet, the site of the transplantation, the viability of the donor cells, the type of vitiligo, the stable period of vitiligo (no isomorphic reaction and depigmentation for more than 6 months), and previous treatment. To minimize the interference of the above factors, we enrolled young patients with stable vitiligo vulgaris who had failed to repigment after traditional drug treatment for more than 1 year and had a stable period for more than 1 year. For each enrolled patient, the skin obtained from the donor site was divided into two parts, one culturing the epithelial sheets under serum- and feeder-free conditions and the other culturing the epithelial sheets under

serum- and feeder-dependent conditions (Fig. 1). After culture, the appearance and thickness of the SFF and SFD epithelial sheets were evaluated by HE staining (Fig. 2a). Both methods resulted in stratified cell skin grafts with epithelioid structures, and there was no significant difference in thickness between the two methods (Fig. 2b). The epithelial sheets obtained by the two methods were transplanted to the patient's skin at the same time. Based on the repigmentation observed in previous patients, a one-month recovery period following grafting was followed by visible repigmentation at approximately 3 months, with the optimal outcome typically achieved within one year. In this study, both methods achieved excellent (full repigmentation) results at the 1-year follow-up after transplantation (Fig. 2c; Supplementary Fig. S1). The kinetics of repigmentation were unable to be examined as a result of the loss of follow-up for certain patients at 1, 3, and 6 months due to unforeseen circumstances. The repigmentation patterns are characterized by uniform diffuse repigmentation, with no significant differences observed between the 2 sheets. This uniform pattern is attributed to ACEG, which replaces the epidermal layer without modifying or stimulating the hair follicles.

### Specific cell types of keratinocytes and fibroblasts in epithelial sheets compared with normal skin revealed by scRNA-seq.
Healthy skin from 3 patients with stable vitiligo was obtained by surgical excision. Its derived epithelial sheets were cultured under SFD and SFF conditions. A total of 9 samples were collected: healthy skin from 3 patients and autologous cultured SFD and SFF epithelial sheets (Supplementary Fig. S2a). Cells dissociated from these 9 samples were subjected to scRNA-seq. After stringent quality controls (Supplementary Figs. S1b–e), transcriptomes of 79200 cells were obtained. A total of 16 distinct cell clusters were identified according to unsupervised hierarchical clustering (Fig. 3a; Supplementary Figs. S1f, 1g). We formed cluster-specific marker genes using differential gene expression analysis to describe the identity of each cell cluster (Fig. 3b; Supplementary Fig. S1h–n). The unbiased cluster identifier was frequently a widely recognized specific cell marker, such as TYRP1, PMEL, and MLANA for melanocytes, KRT14, KRT10, and KRT5 for keratinocytes, and COL1A1, COL1A2, and FN1 for fibroblasts. Notably, we redefined a cluster of fibroblasts using an additional marker called ZNF90 on the basis of established markers for fibroblasts. Therefore, the 16 clusters included 7 clusters of keratinocytes (granular, spinous, LAMB3+basal, KRT15+basal, suprabasal-1, suprabasal-2, proliferation), 2 clusters of fibroblasts (ZNF90+ and DCN+), as well as melanocytes, endothelial cells, T cells, Langerhans cells, mast cells, and neurons (Fig. 3a). Immunofluorescence staining verified the presence of these clusters in epithelial sheets and normal skin. (Fig. 3c; Supplementary Fig. S3).

Changes in cell composition could shed light on how melanocytes maintain their own stability and carry out melanin synthesis. The cell proportions in each cluster were analyzed and exhibited heterogeneity within and among samples (Fig. 3d). Compared with those of healthy skin, the proportions of melanocytes, granular keratinocytes, LAMB3+ basal keratinocytes, and ZNF90+ fibroblasts increased in epithelial sheets cultured under both SFD and SFF conditions (Fig. 3d, e). In a comparison of SFF epithelial sheets to SFD epithelial sheets, the proportions of these 4 cell clusters showed an increase in ZNF90+ fibroblasts and LAMB3+ basal keratinocytes, with a decrease in granular keratinocytes and melanocytes (Fig. 3e). By flow cytometry, we verified that the proportions of LAMB3+ basal keratinocyte clusters and ZNF90+ fibroblast clusters (ESD as a specific marker for ZNF90+ fibroblast cluster, Fig. 3b) increased in SFF epithelial sheets compared to SFD epithelial

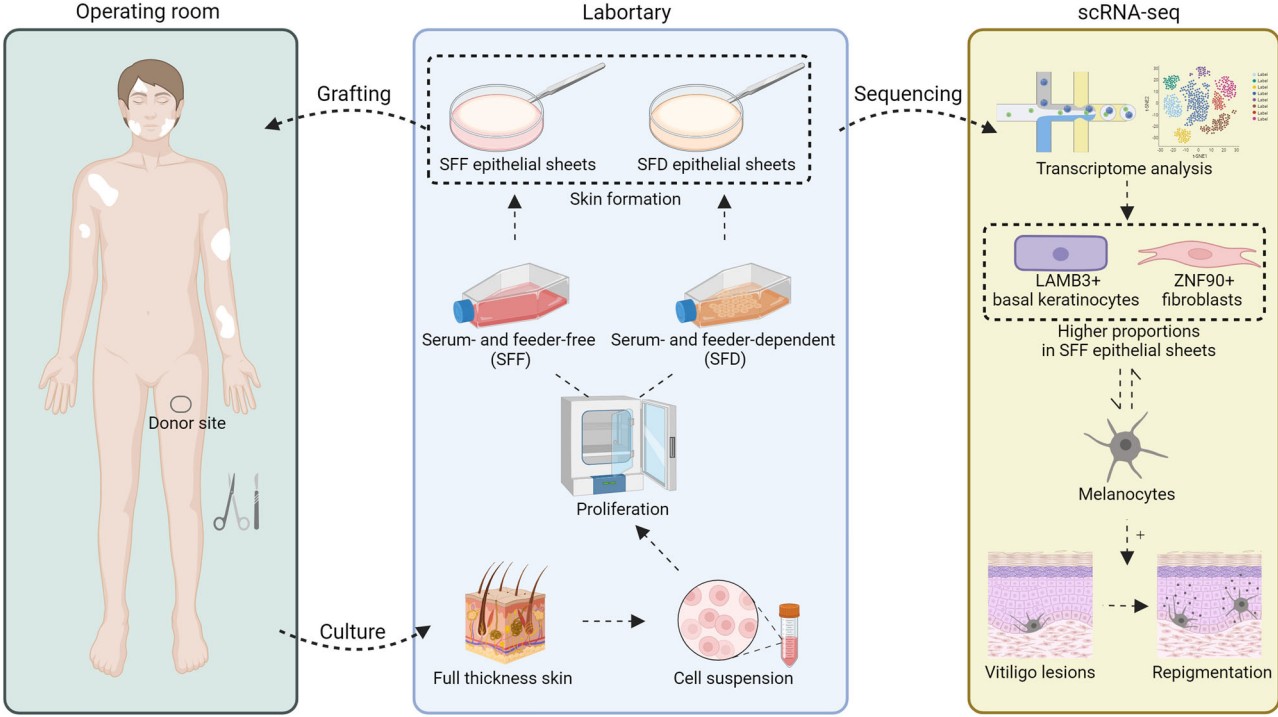

**Fig. 1 The flow diagram of the study.** Schematic overview of autologous cultured epithelial grafting (ACEG) for stable vitiligo, comprising the donor sites acquisition, in vitro cultivation, and grafting. Subsequent single-cell transcriptomic analysis identifying efficacious repigmentation strategies.

sheets (Fig. 3f; Supplementary Fig. S4). Then, we explored the frequencies of these two clusters in scRNA-seq data[14] from vitiligo lesions and healthy skin, and found a decrease in the proportion of LAMB3+ basal keratinocytes in lesions (lesions vs. normal skin: 3.19% vs. 4.32%, $P > 0.05$) and no difference in the proportion of ZNF90+ fibroblasts (lesions vs. normal skin: 1.12% vs. 0.63%, $P < 0.05$) (Supplementary Fig. S5a), which may reflect the correlation between these two clusters of cells and skin pigmentation. Prior research has proposed that KRT6A+ "stress keratinocytes[15]" and Ifngr1+ fibroblasts[14] are implicated in the immune response associated with vitiligo. However, notably, our dataset did not reveal enrichment of Ifngr1+ fibroblasts in both sheets (Supplementary Fig. S5b), compared to healthy skin; stress keratinocytes constituted a small proportion in the SFF epithelial sheets, while a certain proportion of stress 2 subpopulation was observed in the SFD epithelial sheets (Supplementary Fig. S5c). This suggests that the proportion of aberrant cell subpopulations associated with immune response in vitiligo lesions was minor in our SFF epithelial sheets.

In regard to melanocytes, which can directly affect repigmentation after grafting, GSVA (Gene set variation analysis) using Reactome and GO database showed that pathways including "melanin biosynthesis", "PI3K-AKT signaling", "cell-cell adhesion", and "positive regulation of melanocyte differentiation" were enhanced in melanocytes in SFF epithelial sheets compared to SFD epithelial sheets; there was no difference in "melanocyte proliferation" (Fig. 3g; Supplementary Data 1). On the scale of samples, GSEA (Gene set enrichment analysis) of sequenced melanocytes showed no difference between the two epithelial sheets, and enrichment analysis also showed no difference in melanocyte-related pathways (Supplementary Fig. S6; Supplementary Data 1). Thus, melanocytes in SFF sheets may exhibit a higher activity in melanin production and differentiation compared to those in SFD sheets, which could potentially be attributed to the variations observed in cell populations.

**Putative LAMB3+ basal keratinocyte and ZNF90+ fibroblast differentiation trajectories.** While UMAP analysis revealed keratinocyte and fibroblast heterogeneity, we also questioned whether they adhered to particular differentiation trajectories. By ordering keratinocytes in pseudotime, we obtained a triangular trajectory with 2 probable differentiation paths from the vertex to the bottom right and the left corners (Fig. 4a). Differential gene expression analysis showed that LAMB3, which represents basal keratinocytes, and ITGB1, which is related to epithelial stem cell characteristics, were highly expressed at the early stage, while LCE3D/E, which represents granular keratinocytes, and S100A8/A9, which mediates keratinocyte growth inhibition were upregulated at a later stage (Fig. 4b). We investigated the modules of sets of coregulated genes among all 7 keratinocyte clusters and observed that modules 4 and 15 were relatively upregulated in LAMB3+ keratinocytes (Fig. 4c). When these 2 modules were projected to the pseudotime map, they could be found at the right corner at the early stage (Fig. 4d). Next, we conducted pathway enrichment analysis of modules 4 and 15 using Reactome (Fig. 4e) and the GO database (Fig. 4f), and found that laminin interactions, ECM proteoglycans, integrin cell surface interactions, kinase activity, transmembrane signaling receptor activity, and so on were enriched in modules 4 and 15, which suggested that LAMB3+ basal keratinocytes participated in extracellular matrix homeostasis and exhibited vigorous cellular metabolism.

Ordering of fibroblasts in pseudotime arranged them into a hybrid of an initial cyclical trajectory and a subsequent linear trajectory (Fig. 4g). DCN- and COL6A1-expressing cells were mainly dispersed toward the beginning of the trajectory, whereas ZNF90 and HIST3H2A expression levels increased near the end of the trajectory (Fig. 4h). In other words, cell trajectory analysis represented a differentiation pattern from DCN+ to ZNF90+ fibroblasts. Modules 10, 7, 16, 8, and 9 were upregulated in ZNF90+ fibroblasts (Fig. 4i), which was consistent with dynamic changes in feature plots (Fig. 4j). Moreover, pathway enrichment analysis suggested that anaphase transition cell signaling in

**Table 1 The clinical characteristics of the 3 enrolled patients.**

Patients' characteristics

| No. | Age | Gender | Race | Classification | Duration of disease | Stable period | Co-morbidities | Family history | Previous treatment | Outcomes | |
|---|---|---|---|---|---|---|---|---|---|---|---|
| | | | | | | | | | | Effective rate at 1 year | Adverse reactions |
| 1 | 20 s | female | Chinese | Non-segmental | 2 years | 1 year | No | No | TCM*; Topical glucocorticoids | 100% | Pain at the surgery sites |
| 2 | 20 s | female | Chinese | Non-segmental | 5 years | 2 years | No | No | TCM; Topical glucocorticoids; NB-UVB | 100% | No |
| 3 | 30 s | female | Chinese | Non-segmental | 4 years | 2 years | No | No | TCM; Topical glucocorticoids | 100% | Itches at the recipient sites |

ACEG

| Donor Site | Donor area (cm²) | Cell Number 10^6 | Viability | Transplantation Site | Method | K:M | Duration until confluence | Dished (10 cm²) | Total culture Days |
|---|---|---|---|---|---|---|---|---|---|
| Groin | 8.32 (1.6 × 5.2) | 24.4 | 99.50% | Left calf (upper)<br>Left calf (lower) | SFD<br>SFF | 39:1<br>39:1 | 8<br>8 | 20<br>20 | 18<br>18 |
| Groin | 4.5 (1.5 × 3.0) | 35.6 | 95.10% | Right calf<br>Left calf | SFD<br>SFF | 39:1<br>39:1 | 7<br>7 | 16<br>16 | 18<br>18 |
| Groin | 3 (1.0 × 3.0) | 14.2 | 99.00% | Left wrist<br>Right hand | SFD<br>SFF | 35:1<br>35:1 | 8<br>8 | 8<br>8 | 18<br>18 |

*TCM Traditional Chinese Medicine, a hospital-made herbal formulatoin named "Qibai Granules", which includes ingredients such as Astragalus membranaceus (Huangqi) and Angelica dahurica (Baizhi).

mitosis, positive regulation of apoptotic process, keratinization cornified envelope formation, epidermal and epithelial cell differentiation, vasculature development, and so on were enriched in ZNF90+ fibroblasts (Fig. 4k, l). These results demonstrated that ZNF90+ fibroblasts showed more mature differentiation than DCN+ fibroblasts, more importantly, were associated with more developed cellular activity and stronger interactions with epidermal cells and vascular endothelial cells.

**Specific cellular functions of LAMB3+ basal keratinocytes and ZNF90+ fibroblasts.** To elucidate the cellular functions of LAMB3+ basal keratinocytes and ZNF90+ fibroblasts in melanocyte development, we identified DEGs using GSVA via REACTOME and the GO database in 2 groups, LAMB3+ basal keratinocytes and other keratinocytes and ZNF90+ fibroblasts and other fibroblasts.

DEGs were mainly involved with the activation of extracellular matrix (Fig. 5a, b; Supplementary Data 2), including "extracellular matrix organization", "fibronectin matrix formation", "collagen formation", "collagen fibril organization" and so on, when comparing LAMB3+ basal keratinocytes to other keratinocytes. Moreover, upregulated genes were associated with melanocyte proliferation (Fig. 5b). WNT signaling, activating the differentiation of melanocyte precursors[16], was enriched, including "beta-catenin independent WNT signaling", "positive regulation of WNT signaling pathway planar cell polarity pathway", "WNT signaling pathway calcium modulating pathway" and so on (Fig. 5a, b). FGF, an essential growth factor for melanocytes[17], was upregulated via "signaling by FGFR2", "regulation of cell chemotaxis to fibroblast growth factor", and so on (Fig. 5a, b). Upregulated pathways between keratinocyte subtypes (KRT15+ basal keratinocytes, suprabasal keratinocytes 1, suprabasal keratinocytes 2, proliferating keratinocytes, spinous keratinocytes, granular keratinocytes) and other keratinocytes were demonstrated in Supplementary Fig. S7a and Supplementary Data 2.

Regarding fibroblasts, GSVA suggested that pathways involved in collagen, including "assembly of collagen fibrils and other multimeric structures", "intrinsic pathway of fibrin clot formation", "collagen chain trimerization", and "collagen formation", were enriched in ZNF90+ fibroblasts compared to DCN+ fibroblasts (Fig. 5c, d; Supplementary Data 2). These pathways may help fibroblasts to adhere to surrounding cells, such as "positive regulation of cell adhesion mediated by integrin" and "cell junction maintenance", "keratinization", "hair follicle maturation", and "epithelial mesenchymal cell signaling", and so on (Fig. 5c, d). Upregulated pathways between DCN+ fibroblasts and ZNF90+ fibroblasts were demonstrated in Supplementary Fig. S7b and Supplementary Data 2.

**Potential ligand-receptor interactions between LAMB3+ basal keratinocyte and ZNF90+ fibroblast clusters and melanocyte clusters in epithelial sheets.** To characterize the cell–cell communication paradigm between LAMB3+ basal keratinocytes, and ZNF90+ fibroblast clusters and melanocyte clusters in epithelial sheets, we performed an analysis using CellPhoneDB, which contains a repository of ligand–receptor interactions and offers a framework for deducing cell-cell communication networks between two cell types from a single-cell transcriptomics dataset. In the heatmap showing the numbers of interpopulations communications with each other in SFF sheets and SFD sheets (Fig. 6a), we observed more abundant interactions among cells in SFF epithelial sheets in particular, of which LAMB3+ basal keratinocytes displayed the richest interactions with other cell types, including melanocytes.

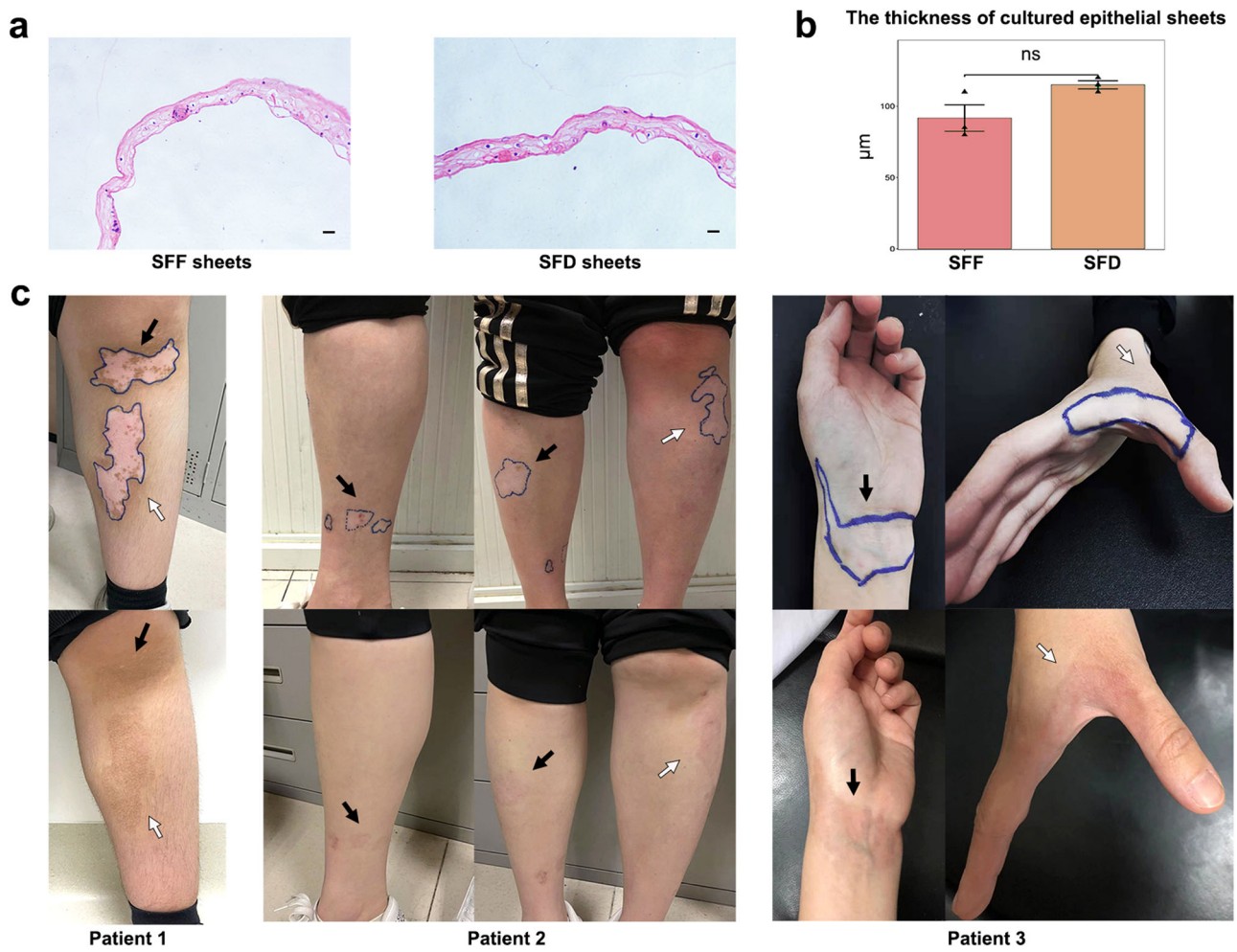

**Fig. 2 ACEG in patients with stable vitiligo. a** HE staining of SFF epithelial sheets and SFD epithelial sheets. **b** Bar chart comparing the thickness of SFF epithelial sheets and SFD epithelial sheets. Scale bar = 100 μm. $n = 3$ cultured epithelial sheets per group analyzed with unpaired Student's $t$-test ($P = 0.12$). Data presented as mean ± SD. **c** Digital photographs of the vitiligo lesions of 3 patients at baseline (upper) and at the 1-year follow-up (lower). The black arrows point to the site receiving SFD epithelial sheets grafting, and the white arrows point to the site receiving SFF epithelial sheets grafting.

Next, we established ligand–receptor pairs between LAMB3+ basal keratinocytes and melanocytes in SFF versus SFD sheets (Supplementary Data 3). Significantly altered signals from LAMB3+ basal keratinocytes associated with melanocyte homeostasis including JAG, NOTCH and EGF signaling[18] were observed in both SFF and SFD sheets, whereas more interactions were involved in SFF sheets (Fig. 6b). For example, EGFR and its receptors including MIF, GRN, and COPA were upregulated in the interactions between LAMB3+ basal and melanocytes from both sheets, while JAG1-NOTCH2/3/4 and NOTCH1-NOV ligand–receptor pairs were significantly altered only in SFF sheets. Additionally, we investigated variations in ligand signals transmitted by melanocytes. In both sheets, CD44, which facilitates cell adhesion by hyaluronic acid[19], and CD46, which protects melanocytes against complement-mediated damage[20], were highly expressed in melanocytes, thereby activating LAMB3+ keratinocytes via several patterns such as CD44-FGFR2, CD44-HBEGF, and CD46-JAG1 pairs (Fig. 6b). Notably, KIT-KITLG pairs that play an important role in melanocyte migration and survival[21], and NRP2-VEGFA/SEMA3C pairs that influence cell migration and angiogenesis[22] were significantly altered only in SFF sheets. Representative ligand–receptor circle figures also indicated that JAG, NOTCH, and CD44 signaling interactions increased between 2 cells in SFF sheets compared to SFD sheets (Fig. 6c).

Moreover, ligand–receptor pairs between ZNF90+ fibroblasts and melanocytes were investigated (Supplementary Data 3). JAG1-NOTCH, NRP2-VEGFA/SEMA3C, and MIF-TNFRSF14 pairs were significantly altered in SFF sheets where melanocytes express receptors and respond to ligand signals from ZNF90+ fibroblasts (Fig. 6b, left), while only MIF-TNFRSF14 pairs that might induce hypopigmentation[23] to maintain melanocyte homeostasis were dramatically changed in SFD sheets (Fig. 6b, right). Moreover, ligand signals that were significantly overexpressed in melanocytes including NRP2, KIT, CD44, and CD46 were accepted by ZNF90+ fibroblasts from SFF sheets (Fig. 6b, left), while only KIT signaling was enhanced in melanocytes from SFD sheets (Fig. 6b, right). These results were also supported in the circle figures (Fig. 6c). Taken together, more interactions between LAMB3+ basal keratinocyte and ZNF90+ fibroblast clusters and melanocyte clusters were observed in SFF sheets than in SFD sheets.

To interpret the interaction between these cells, we conducted further validation. Immunofluorescence staining demonstrated that these two clusters co-localized with melanocytes in SFF, SFD epithelial sheets and healthy skin (Fig. 6e). Keratinocytes and fibroblasts were isolated through primary culture, and qPCR revealed that knockdown of *LAMB3* in keratinocytes and knockdown of *ZNF90* in fibroblasts resulted in downregulation of ligand-receptor genes associated with melanocytes (Fig. 6f; Supplementary Fig. S8).

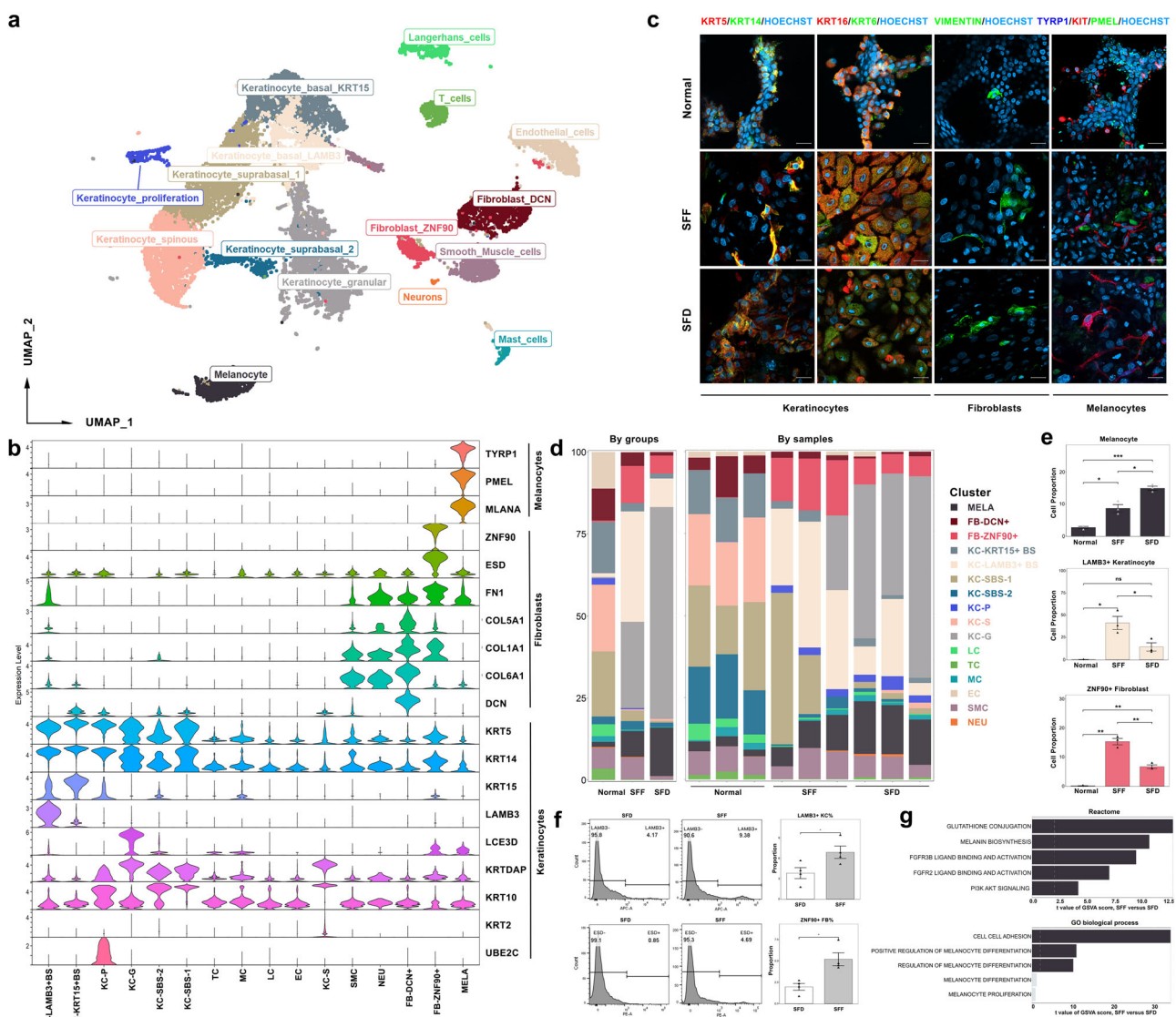

**Fig. 3 Cell heterogeneity of normal skin and its derived cultured epithelium under SFF and SFD conditions. a** Uniform manifold approximation and projection (UMAP) plot of 10X genomics-based single cells identifying 16 major cell types derived from 3 normal skin, 3 SFF epithelium, and 3 SFD epithelium samples. **b** Violin plots showing the expression of specific marker genes for keratinocytes, fibroblasts, and melanocytes in each subpopulation. **c** Immunofluorescence staining of KRT5 (red), KRT14 (green), KRT16 (red), and KRT6 (green) for keratinocytes, VIMENTIN (green) for fibroblasts, TYRP1 (dark blue), KIT (red), and PMEL (green) for melanocytes in SFF, SFD epithelium and normal skin (Hoechst with light blue for nucleus). Scale bar = 50 μm. **d, e** Cell proportions of each cluster presented by groups and samples (**d**), melanocyte cluster (normal vs. SFF: $P = 0.028$, normal vs. SFD: $P = 0.00046$, SFF vs. SFD: $P = 0.017$), LAMB3+ basal keratinocyte cluster (normal vs. SFF: $P = 0.031$, normal vs. SFD: $P = 0.092$, SFF vs. SFD: $P = 0.046$), and ZNF90+ fibroblast cluster (normal vs. SFF: $P = 0.0051$, normal vs. SFD: $P = 0.0088$, SFF vs. SFD: $P = 0.0052$) (**e**) in normal skin, SFF epithelium, and SFD epithelium. **f** Flow cytometry showing the proportion of LAMB3+ keratinocytes ($P = 0.041$) and ZNF90+ fibroblasts ($P = 0.015$) in SFF and SFD epithelial sheets. **g** GSVA analysis of melanocytes between SFF and SFD epithelial sheets. Shown are t values from a linear model. For flow cytometry, $n = 4$ independent experiments over cultured epithelial sheets from two additional recruited patients. Data from (**e, f**) presented as the mean ± SD, analyzed with unpaired Student's t-test. (ns > 0.05, *$P < 0.05$, **$P < 0.01$, ***$P < 0.001$). MELA, Melanocytes; FB-DCN+, DCN fibroblasts; FB-ZNF90+, ZNF90 fibroblasts; KC-KRT15 + BS, KRT15 basal keratinocytes; KC-LAMB3 + BS, LAMB3 basal keratinocytes; KC-SBS-1, suprabasal-1 keratinocytes; KC-SBS-2, suprabasal-2 keratinocytes; KC-P, proliferating keratinocytes; KC-S, spinous keratinocytes; KC-G, granular keratinocytes; LC, Langerhans cells; TC, T cells; MC, mast cells; EC, endothelial cells; SMC, smooth muscle cells; NEU, neurons.

## Discussion

Although surgical interventions including autologous cell and tissue grafting for patients with stable vitiligo have been widely explored[1], studies on autologous cultured epithelial grafting have stagnated, and the underlying mechanisms leading to repigmentation remain unexplained. ACEG, most likely a combination of cell grafting and tissue grafting, requires fewer donor sites to achieve a more extensive and uniform repigmentaion. Notably, our modified autologous cultured epithelial sheets without serum and feeder, which showed no differences in repigmentaion compared to Rheinwald and Green's approach, according to the case-control study by the authors[24], is not only stable and replicable, but also a secure and effective approach for treating stable vitiligo. In this study, we confirmed again that ACEG with SFF and SFD sheets is comparable in terms of surgical complexity, clinical effectiveness, and adverse effects.

The serum and feeder of the SFD-conditioned sheets can support the formation of the graft, while the SFF sheets exhibit

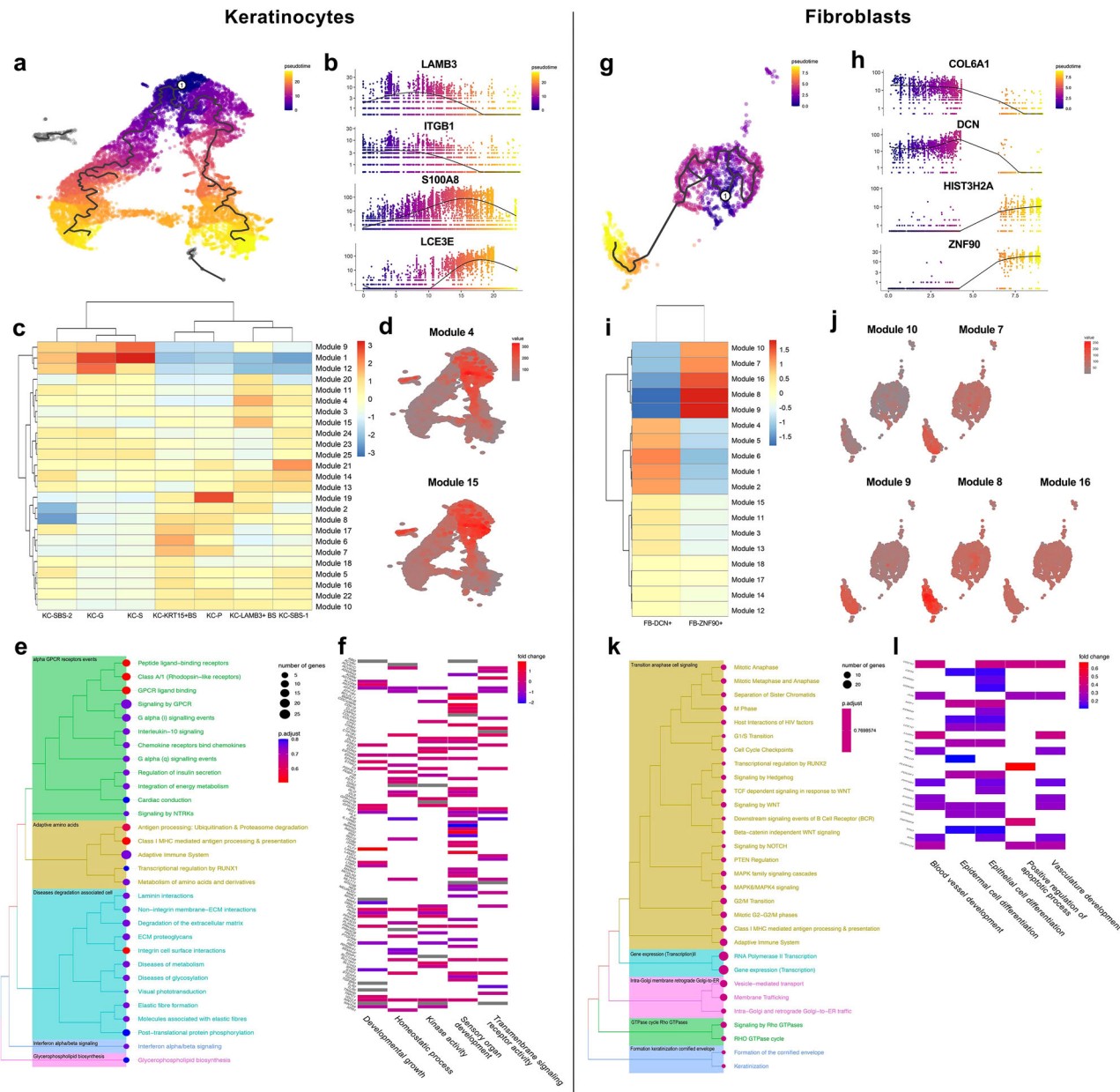

**Fig. 4 Differentiation characteristics in pseudotime of LAMB3+ basal keratinocytes and ZNF90+ fibroblasts. a, g** Pseudotime trajectory map of keratinocytes (**a**) and fibroblasts (**g**). **b, h** Featured plot genes in pseudotime of keratinocytes (**b**) and fibroblasts (**h**) upregulated at the early or later stage. **c, i** Gene modules among keratinocyte (**c**) and fibroblast (**i**) subpopulations. Each module represented a set of co-regulated genes. **d, j** Diffusion map showing the expression levels of LAMB3+ basal keratinocytes (**d**), and ZNF90+ fibroblasts (**j**) within the specific region of the trajectory. **e, k** Tree plot of enriched pathways of modules 4 and 15 (**e**) and modules 10, 7, 9, 8 and 16 (**j**) using Reactome pathway databases. **f, l** Heatmap plot of enriched pathways of modules 4 and 15 (**f**) and modules 10, 7, 9, 8 and 16 (**l**) in GO databases.

potential advantages in safety. However, the thickness of the SFD- and SFF- conditioned sheets and the efficacy between the two methods showed no significant difference. Therefore, a possible explanation for this might be that some other elements assist melanocytes in achieving repigmentation in SFF sheets. Here, we provide a molecular characterization of the different cell types observed in cultured epithelial sheets derived from healthy skin in vitiligo patients using scRNA-seq. We identified 16 cell types, almost all of which had previously been described[13,25]. Considering that cellular transcriptomes can be affected under different culture conditions with different compositions of medium, we compared the cell compositions of cultured epithelium sheets with those of healthy skin. One of the findings was a decrease in the proportion of immune cells in the SFD and SFF sheets, which

may be due to the absence of stimulators in the culture medium to maintain the growth of immune cells. However, this low immunogenicity favors transplantation. A further discovery is that the SFF sheets were found to contain a significantly higher percentage of LAMB3+ basal keratinocytes and ZNF90+ fibroblasts, and the subsequent analysis may support the hypothesis that these 2 clusters facilitate the function of melanocytes on the basis of the lower cell proportion of melanocytes in SFF sheets compared to SFD sheets.

LAMB3+ basal keratinocytes play an important role in the formation of extracellular matrix, supported by the pathway enrichment analysis in pseudotime and the GSVA results. Previous findings suggested that the LAMB3 gene family is involved in cell proliferation, colony formation, and keratinocyte

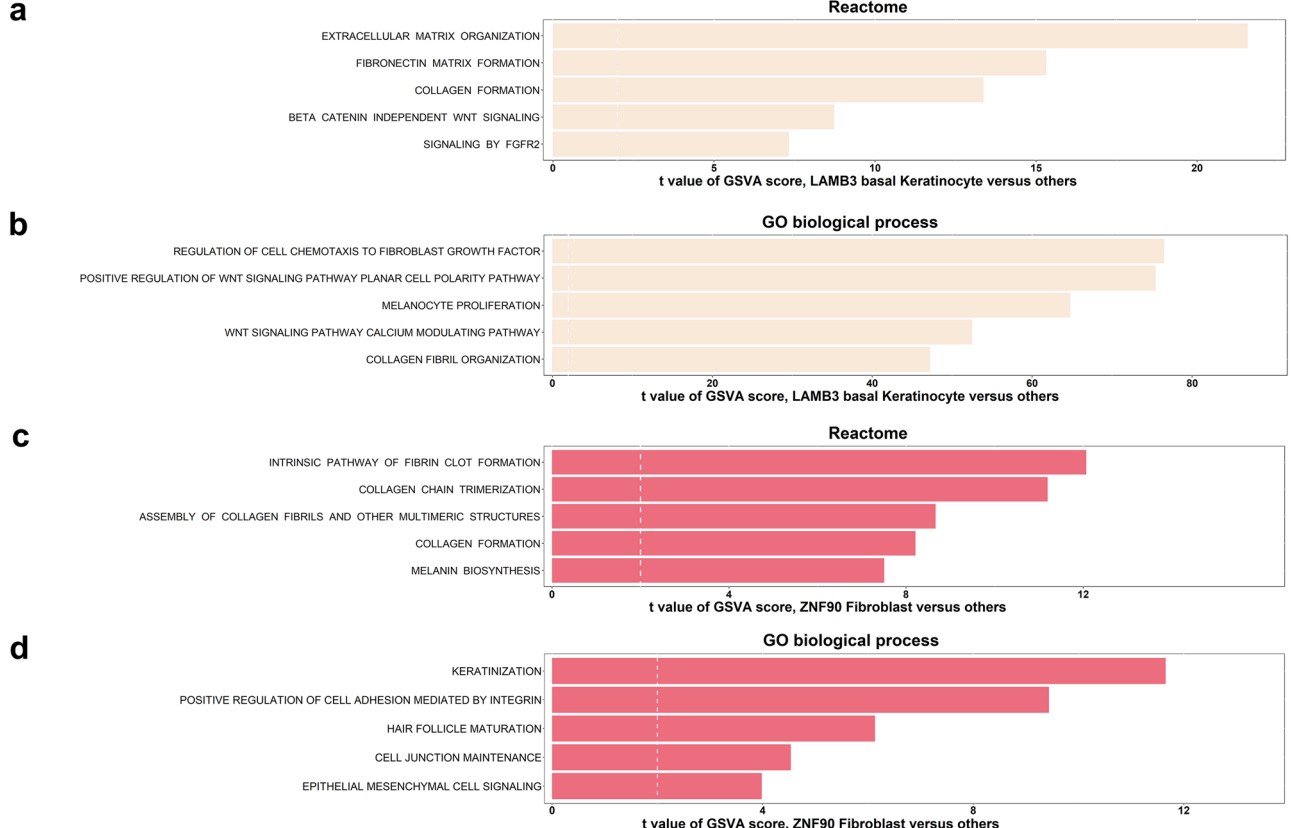

**Fig. 5 GSVA analysis of LAMB3+ basal keratinocytes and ZNF90+ fibroblasts. a, b** Bar plot depicting the upregulated Reactome (**a**) and GO (**b**) pathways between LAMB3+ basal keratinocytes and other keratinocytes. **c, d** Bar plot showing the upregulated Reactome (**c**) and GO (**d**) pathways between ZNF90+ fibroblasts and other fibroblasts. Shown are *t* values from a linear model.

adhesion[26]. Its mutations impair the functions of laminin 332, which are linked to epidermal-dermal adhesion and may cause junctional epidermolysis bullosa[27,28]. More significantly, laminin-332 promoted adhesion and migration in melanocytes, as well as melanin synthesis by increasing intracellular levels of tyrosine in melanocytes[29–31]. It has been reported that in vitro cultures with laminin coating could facilitate the expansion of melanocytes[32,33]. Therefore, LAMB3 and laminin-332 potentially act as driving regulators and predictive biomarkers for melanocyte growth and vitiligo progression.

Cell-cell communication analysis revealed that JAG1-NOTCH2/3/4 pairs and EGFR-MIF/GRN/COPA pairs broadcast by LAMB3+ basal keratinocytes and accepted by melanocytes, were abundant in SFF sheets. Notch signaling has been identified as a trigger for keratinocyte differentiation and a key component in the interaction between melanocytes and surrounding keratinocytes[34–36]. Although initially found in skin's nonimmune cells, MIF (macrophage migration inhibitory factor), GRN (granulin precursor), and COPA (COPI coat complex subunit alpha) were associated with tumorigenesis and inflammatory diseases[37–39]. Additionally, the interaction between CD44 and members of the EGFR and FGFR family of receptor tyrosine kinases (RTKs) on melanocytes to LAMB3+ keratinocytes was upregulated. It has been demonstrated that CD44-FGFR is needed in the interactions between epithelial cells in the apical ectodermal ridge and mesenchymal cells in vertebrate limb development;[40] CD44 is involved in the maturation of HBEGF and the subsequent activation of ErbB4 to regulate physiological tissue remodeling[41]. Therefore, we can infer that LAMB3+ basal keratinocytes may enhance repigmentation by exerting an influence on the skin microenvironment within the whole epidermal

melanin unit, and signaling on melanocytes can also deliver positive feedback in this setting.

Another discovery was that ZNF90+ fibroblasts constituted a later stage of fibroblast differentiation and were associated with collagen formation and melanin biosynthesis, according to differentiation trajectories and GSVA results. ZNF90 belongs to the zinc finger protein family that is known to be involved in transcriptional regulation by binding specifically to short DNA regions and is implicated in the differentiation of keratinocytes and the development of skin diseases, such as psoriasis[42]. Further investigations are needed to determine its role in fibroblasts and corresponding interactions with melanocytes.

Next, cell-cell communication analysis revealed that JAG1-NOTCH signaling, NRP2-VEGFA/SEMA3C signaling and CD44-HBEGF signaling were significantly altered between ZNF90+ fibroblasts and melanocytes in SFF sheets. First, JAG1-NOTCH signaling is involved in skin epithelial-to-mesenchymal transition, hair follicle maintenance, and wound healing[43–45], in addition to melanocytic proliferation[18], which also matches our GSVA results of ZNF90+ fibroblasts. Second, NRP2 (neuropilin 2), a transmembrane glycoprotein, functions as neuronal guidance through binding class 3 semaphorins (SEMAs), which are essential for neural crest cells migrating dorsolaterally to form melanocytes[22]. NRP2 also participates in angiogenesis by interacting with vascular endothelial growth factor (VEGF)[46–48], which is suitable for rehabilitating the recipient skin. Third, CD44 was reported to play a pivotal role in recruiting fibroblasts to injury sites and coordinating fibroblast behaviors in tissue remodeling[49–51]. Therefore, our findings shed light on the signaling pathways between ZNF90+ fibroblasts and melanocytes that are mainly beneficial for repair and regeneration after

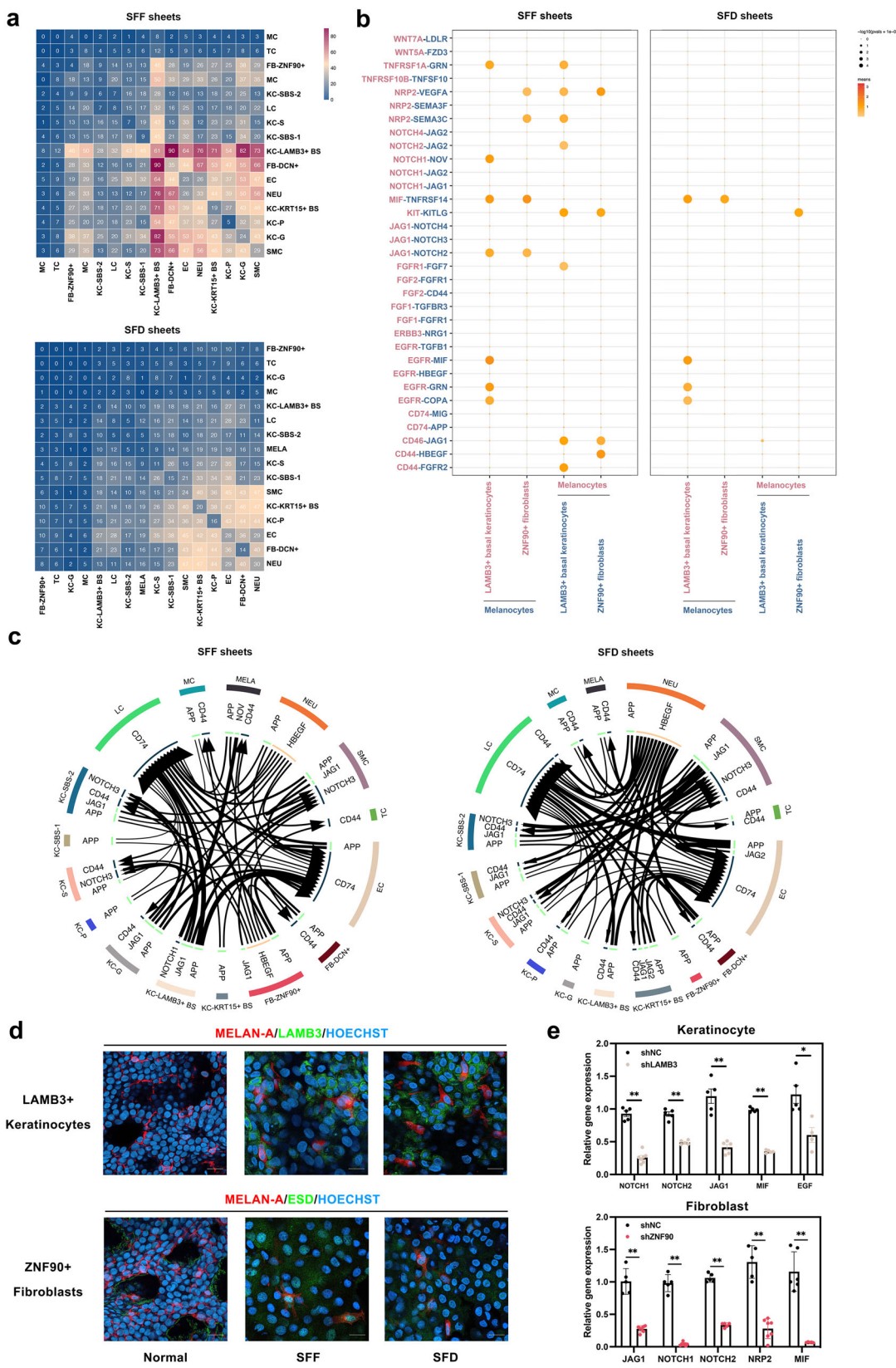

transcription. It can thus be suggested that ZNF90 may be a viable cellular target of fibroblasts in the development of vitiligo.

Moreover, the expression of fibronectin 1 (FN1) was significantly higher in the ZNF90+ fibroblasts cluster than in DCN+ fibroblast cluster (Fig. 3b). Fibronectin, a member of the glycoprotein family, has been associated with the migration and

adhesion of melanocytes by various receptor and ligand target sequences[52]. Fibronectin-coated culture plates are widely used to generate melanocytes from human embryonic stem cells and induced pluripotent stem cells[53,54]. Notably, FN1 has also been identified as an extracellular matrix component of melanocyte extracellular vesicles after UVB radiation[55], suggesting that it can

**Fig. 6 Potential ligand–receptor interaction analysis in LAMB3+ basal keratinocytes and ZNF90+ fibroblasts. a** Heatmap illustrating the numbers of interclusters communications with each other in SFF epithelial sheets (upper) and in SFD epithelial sheets (lower) versus normal skin. **b** Dot plot exhibiting the ligand–receptor pairs with significant changes between LAMB3+ basal keratinocyte and ZNF90+ fibroblast clusters and melanocyte cluster in SFF epithelial sheets (left) and in SFD epithelial sheets (right). The pink letter denotes ligands (vertical ordinate) and cell types releasing ligand signals (horizontal ordinate), and the blue letter denotes receptors (vertical ordinate) and cell types expressing receptors and receiving ligand signals from other clusters (horizontal ordinate). **c** Network diagram representing the relative signaling mentioned in the dot plot within LAMB3+ basal keratinocyte and ZNF90+ fibroblast clusters and the melanocyte cluster in SFF epithelial sheets (left) and in SFD epithelial sheets (right). All arrows are pointing to the receptors. **d** Immunofluorescence staining of co-localization of LAMB3+ keratinocytes and ZNF90+ fibroblasts with melanocytes in SFF, SFD epithelium and normal skin. MELAN-A (red) was for melanocytes, LAMB3 (green) was for LAMB3+ keratinocytes, and ESD (green) was for ZNF90+ fibroblasts (Hoechst with light blue for nucleus). Scale bar = 50 μm. **e** Expression levels of *JAG1* ($P = 0.008$), *NOTCH1* ($P = 0.0025$), *NOTCH2* ($P = 0.0008$), *MIF* ($P = 0.008$), and *EGF* ($P = 0.016$) in keratinocytes for the *LAMB3* knockdown group (shLAMB3) and the control group (ShNC), as well as the expression levels of *JAG1* ($P = 0.0025$), *MIF* ($P = 0.0043$), *NOTCH1* ($P = 0.0025$), *NOTCH2* ($P = 0.0079$), *NRP2* ($P = 0.0025$) in fibroblasts for the *ZNF90* knockdown group (shZNF90) and the control group (ShNC). For qPCR, $n = 4$–7 independent experiments conducted over two biologically independent patient-derived samples. Data presented as the mean ± SD, analyzed with unpaired, nonparametric *t*-test. (*$P < 0.05$, **$P < 0.01$).

maintain melanocyte homeostasis in pathological conditions. Thus, it could be hypothesized that FN1 is an essential molecular marker for ZNF90+ fibroblasts, which facilitates repigmentation when transplanting cultured epithelial sheets onto patients with vitiligo.

In conclusion, our research contributes to the growing body of literature on single-cell RNA sequencing for vitiligo. According to the authors' understanding, the present work is the first to identify potential cellular targets for vitiligo repigmentation by surgical interventions. From bench to bedside and back again, we found that, in the absence of serum and feeder, LAMB3+ basal keratinocytes and ZNF90+ fibroblasts may promote repigmentation in cultured autologous epithelial sheets in the treatment of patients with refractory vitiligo. Culture conditions, while strongly influencing the properties of the cell composition in the sheets, do not appear to affect the clinical efficacy of repigmentation. This suggests that SFF sheets exhibit similar efficiency to SFD sheets while offering enhanced safety, marking a promising avenue for further clinical application.

## Methods

**Study design.** For this study, patients with vitiligo who visited Fudan University Huashan Hospital's outpatient clinic between November 1, 2015 and June 30, 2022 were eligible. Patients aged 14–85 years, male or female, with stable vitiligo (defined as no progression of existing lesions, no appearance of new lesions, and no Koebner phenomenon for at least 12 months), an inadequate response to a variety of medical treatments for vitiligo, and willingness and ability to undergo treatment with ACEG under the two types of sheets could be included. Patients who decided not to receive transplantation of ACEG for the treatment of stable vitiligo and patients with unwillingness to collect the sample for scientific research, private area lesions, extensive depigmented areas (defined as an area covering more than 1% of body surface area), serious systemic diseases, keloid diathesis, or infections were excluded.

This study was approved by the Ethics Committee of Huashan Hospital, Fudan University (KY2020-698, KY2020-1137) and registered with chictr.org.cn (ChiCTR2100051405). All participants provided written informed consent. All ethical regulations relevant to human research participants were followed. Details of the clinical research design, the approval by the Ethics Committee, and informed consent can be found in the Supplementary Data 4.

**ACEG procedures.** ACEG procedures include donor sites acquisition, epithelial sheets culture, surgery, postoperative care, and follow-up. Donor skin was collected as a full-thickness skin biopsy specimen (3–6 cm$^2$) from a pigmented site, typically the groin, and dissociated with 0.5% Type I collagenase (Gibco, Grand Island, NY, USA).

Two cultivation methods were utilized for each patient. SFF epithelial sheets: The cell mixture was grown in specified keratinocyte serum-free medium (basal medium without bovine pituitary extract including insulin, epidermal growth factor (EGF), fibroblast growth factor (FGF), and 0.1 mM Ca2+) (Gibco) in 5% $CO_2$ at 37 °C until confluence. SFD epithelial sheets: Cells were grown with an irradiated mouse Swiss 3T3 (i3T3) fibroblast feeder layer (20,000 i3T3 per cm$^2$ irradiated at 6000 rad to prevent further proliferation) in medium containing 75% Dulbecco's modified Eagle's medium (Invitrogen, Oakville, ON, Canada) and 25% Ham's F12 medium (Invitrogen) with 5% fetal clone II serum (HyClone, Logan, UT), 10 ng/mL EGF, 24.3 mg/mL adenine, 5 mg/mL insulin, 0.4 mg/mL hydrocortisone, 0.212 mg/mL isoproterenol, 100 IU/mL penicillin G, and 25 mg/mL gentamicin. Then, the culture medium was changed to keratinocyte growth medium (1.0 to 1.4 mM Ca$^{2+}$) and incubated for 7 to 10 days. The Ca$^{2+}$ concentration was 0.15 mM on Day 1, 0.5 mM on Days 2 to 3, 1.0 mM on Days 4 to 5, and 1.4 mM on Days 6 to 7 and even until Day 10. For both methods, cells were transplanted to a 10 × 10 cm hyaluronic acid sheet on the bottom of a similar-sized dish. Cells continued to proliferate and created a keratinizing multiramified tissue that resembled the organoid structure of an epithelium containing keratinocytes and melanocytes. The epithelium was transplantable after 14 to 21 days. DOPA staining determined the culture's melanocyte-keratinocyte ratio. HE staining was performed to assess the microscopic morphology of the skin sheet.

Before surgery, vitiligo lesions were imaged and assessed by Wood's lamp and reflectance confocal microscopy. For semi-automatic image analysis, surgical marking pens were used to outline recipient locations. Before grafting, recipient areas were de-epithelialized through dermabrasion. They were washed with sterile saline and coated with cultivated epithelial autografts. Grafts were fixed and immobilized by dressings made of UrgoTui (LABORATORIES URGO), dry gauze, and bandages. After one week, all dressings were replaced except the Adaptic gauze with silicon N/A gauze bandages (Johnson & Johnson Medical Ltd., Ascot, UK), which protected the wound for another week. Bandages were removed 7 to 14 days later to expose the repigmented epithelialized surface. Patients were instructed not to sunburn after surgery. After surgery, 5 to 10 min of indirect sun exposure was allowed daily for 1 month. This sun exposure method allowed the graft sites to repigment without darkening.

Digital photos were taken before and after surgery. After transplantation, follow-up was conducted at 1, 3, 6, and 12 months. Repigmentation rate was primarily assessed by two dermatologists through independent visual examination, with Wood's lamp and reflectance confocal microscopy serving as auxiliary diagnostic tools. Subsequently, the overall effective rate

was determined and was classified as poor (<25% lesion repigmentation), fair (26%–50%), good (51%–75%), and excellent (>75%). Adverse reactions were recorded.

**Single cell transcriptome sequencing.** The single cell cDNA library was created by Chromium Next GEM Single Cell 3' Reagent Kits v3.1 and 10X Genomic Chromium following the manufacturer's instructions. For each sample, we aim to capture 5000 ~ 10000 cells. Transcriptome sequencing was performed on the Illumina NovaSeq 6000 platform with the PE150 strategy, and approximately 500 M data were generated for each sample. Details of the sequencing method can be found in the Supplementary Data 5. Briefly, Cell Ranger performs data quality statistics on the raw data and compares the reference genomes to the Ensembl database. R software (4.1.3) and Python (3.8.4) were used for the bioinformatic analysis. Details of these analyses can be found in the Supplementary Data 6.

The harmony1(https://github.com/immunogenomics/harmony) package of R software was used for the integration of single-cell data and removal of batch effects. The Seurat (4.0.6, https://satijalab.org/seurat/) package of R software was used for clustering analysis. ClusterTree (0.5.0) was used to determine the resolution for clustering analysis. A standard Seurat SCTransform workflow (resolution = 0.3, reduction = PCA, dims = 1:40) was performed for clustering analysis. For integrative analysis, we obtained data from healthy skin and lesional skin from patients with vitiligo[14] and the proportion of LAMB3+ keratinocytes and ZNF90+ fibroblasts. Then, the enrichment of keratinocyte[15] and fibroblast[14] subsets associated with immune cell recruitment and suppressing repigmentation was examined in our dataset.

The cell trajectory analysis was performed by the Monocle3 (https://cole-trapnell-lab.github.io/monocle3/) package of R software (4.1.3). The GSVA3 package of R software (4.1.3) was used to perform gene set enrichment using the Canonical pathways database (v7.5.1, https://www.gsea-msigdb.org/gsea/msigdb/collections.jsp#C3) and ontology gene set (https://www.gsea-msigdb.org/gsea/msigdb/human/collection_details.jsp#C5), and the GSVA results were further compared by the Limma (3.15) package. For cell-cell communication analysis, the cell curated receptors, ligands and their interactions were analyzed by the CellphoneDB (v4, https://github.com/ventolab/CellphoneDB) python package.

**Immunofluorescence.** For immunocytochemistry analysis, samples were fixed in 4% paraformaldehyde (15 min, room temperature). The cells were washed with phosphate buffer solution (PBS) three times and fixed in 1% BSA PBST (30 min, room temperature). Primary antibodies were incubated overnight at 4 °C in blocking buffer. Samples were stained with the species-specific fluorophore-conjugated secondary antibody (1 h, room temperature); nuclei were visualized with Hoechst. Samples were observed with Zeiss LSM 800 confocal microscope and analyzed with Zen software 2.3. Three independent experiments were performed for each sample. The list of antibodies can be found in the Supplementary Table. S1.

**Flow Cytometry.** Cell suspensions were prepared by digesting epidermal sheets with dispase II (STEMCELL, Canada) to achieve concentrations exceeding $1 \times 10^6$/ml. Intracellular staining used the eBioscience™ Intracellular Fixation & Permeabilization Buffer Set (Thermo Fisher Scientific, 88-8824-00). After incubation at room temperature for 30–60 min, cells were centrifuged at 500 g for 5 min, and the pellet was resuspended in 100 µl of Permeabilization Buffer. Fluorescent antibodies and DAPI were added, and cells were incubated for at least 30 min at room temperature. After centrifugation and resuspension in

Permeabilization Buffer (500 g, 5 min), samples were analyzed using BD LSRFortessa with image processing via FlowJo version 10.4.0. Each experiment was performed thrice.

**Cell transfection.** Primary keratinocytes and fibroblasts were isolated from normal skin of patients with vitiligo and cultured separately in Defined Keratinocyte SFM (Thermo Fisher Scientific) and Fibroblasts Medium (ScienCell) to expand keratinocytes and fibroblasts. The lentivirus vectors *LV-shRNA-LAMB3*, *LV-shRNA-ZNF90* and *LV-shRNA-NC* were purchased from Genepharma. Keratinocytes and fibroblasts were infected with lentiviral suspensions containing *LAMB3* and *ZNF90*, respectively, at an MOI of 20. After 12 h, the culture medium was replaced, and complete culture medium was added. Transfection efficiency was observed under a confocal microscope after 72 h. The culture medium was then replaced with medium containing 1–2 µg/ml puromycin (Thermo Fisher Scientific) for one week of resistance selection, resulting in *LAMB3*-knockout keratinocytes and *ZNF90*-knockout fibroblasts.

**qPCR (quantitative polymerase chain reaction).** The relevant target genes were tested in the transfected shLAMB3 keratinocytes and shZNF90 fibroblasts to validate the knockout. Genes associated with ligand-receptor pairs involved in cell-cell communication with melanocytes were also examined. RNA extraction from keratinocytes and fibroblasts in all groups was performed using the ES Science RNA Fast Extraction Kit (RN001). Reverse transcription of RNA from each group of cells was carried out using the ES Science Fast Reverse Transcription Kit (RT001). qPCR was conducted using ES Science 2X Super SYBR Green qRT-PCR Master Mix (QP002). All procedures were performed according to the manufacturer's instructions. The primers listed in Supplementary Table 2 were used to assess gene expression levels, including *LAMB3, ZNF90, JAG1, NOTCH1, NOTCH2, MIF, EGF,* and *NRP2*. The expression levels of all genes were normalized to GAPDH (for gene expression related to ECM and adhesion molecules). Relative gene expression levels were determined using the 2^-ΔΔCt method in at least three independent experiments.

**Statistics and reproducibility.** All experiments were repeated at least three times. Statistical analyses were performed using R software and Prism 10. Data represent the mean ± standard deviation (SD). A two-tailed, unpaired Student's t-test was conducted to compare the values between subgroups for quantitative data. $P < 0.05$ was considered to be statistically significant.

**Reporting summary.** Further information on research design is available in the Nature Portfolio Reporting Summary linked to this article.

## Data availability

The gene sequencing data in this article have been deposited at https://www.ncbi.nlm.nih.gov/ as BioProject: PRJNA802332. All data, codes, and materials in the analysis can be provided to any researcher for the purposes of reproducing or extending the analysis by contact with the corresponding author. Source data are available in Supplementary Data 7.

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

## Acknowledgements

This work has been supported by the National Natural Science Foundation of China (82073465, 82273562, 81671550), the Shanghai Association for Science and Technology (21140900800, 20Y11905500), and the grants from the Shanghai Engineering Research Center of Hair Medicine (19DZ2250500), Shanghai Municipal Commission of Health and Family Planning (No. 2023ZZ02018), Shanghai Municipal Key Clinical Specialty (No-shslczdzk01002), Ministry of Science and Technology of China (2018YFA0107900),

Shanghai Municipal Government (2019CXJQ01), and Peak Disciplines (Type IV) of Institutions of Higher Leaning in Shanghai. We sincerely thank the doctors, nurses, and laboratory staff in our department and institute for their contribution to this study. We are grateful to all patients for participation.

## Author contributions

W.L., X.Z., J.L., and Q.Z.* (Qing Zang) are joint first authors. Q.Z.# (Q.Z.), J.L., and J.X. obtained funding. J.X., X.Z., Q.Z.#, Y.L., and S.C. designed the study. W.L., X.Z., Q.Z.*, J.L., S.C., S.H., Q.Z.#, and J.S. collected the data. X.Z., W.L., J.L., Q.Z.*, S.C., and Y.X. were involved in data cleaning, mortality follow-up, and verification. X.Z., W.L., Q.Z.#, and J.X. analyzed the data. W.L., Q.Z.#, and X.Z. drafted the manuscript. X.Z., W.L., T.J., Q.Z.#, F.W., and J.X. contributed to the interpretation of the results and critical revision of the manuscript for important intellectual content and approved the final version of the manuscript. All authors have read and approved the final manuscript.

## Competing interests

The authors declare no competing interests.
