## [Peer review file · Communications Biology]

Reviewers' comments:

Reviewer #1 (Remarks to the Author):

The manuscript is a well-constructed study with valuable information that will add to the related literature

Here are my comments

1. line 94: vitiligo vulgaris Not vulgaris vitiligo
2. line 110: The 9 samples; please identify their details
3. line 118: makers? Do you mean markers
4. line 238: please add details on your previous study
5. line 366: please mention how de-epithelization was performed
6. Please add Wood's light photos of recipient sites
7. Please add a table summarizing patients' demographic and clinical data
8. The paper needs professional English editing

Reviewer #2 (Remarks to the Author):

The manuscript by Jinhua Xu and coauthors contains valuable and unique human samples and datasets, with potential value for clinical translation and biological discovery. The paper contains novel results and could be of significant interest for scientific community beyond the field of vitiligo studies.

My major concern is that the currently presented experimental results do not provide enough supportive evidence for the conclusions of the manuscript since the data are mainly descriptive. Further experimental evidence and expansion of clinical data would be beneficial to further support the central hypothesis and conclusions. The manuscript would benefit from providing more methodological details to facilitate the reproduction of experiments by other scientists.

Major concerns:

1. In the provided informed consent template form (supplementary data), patient consented to the collection, processing, analysing and publishing their anonymized nonidentifiable clinical and demographic data. Could the authors provide also the informed consent form templates for tissue analysis by scRNA-seq (where SNPs could be identifiable) and other experiments or describe in manuscript how these studies were regulated.

2. The graphic representation of scRNA-seq data (QC and results) should be improved. The authors should report cell number contribution of each experimental condition (SFF, SFD and healthy skin) to the total integrated dataset, please see as example also Alivernini S. et al, Nat Med 2020, Extended data Fig 2a). QC of scRNA-seq data should be shown per each sequenced sample (n=9), and the same is true for the results in Fig 3. In this way, the cell type variability across and within treatment groups will be fully apparent, please see as example also Figure 2b,c and Suppl Fig. 1a in Edalat S et al, bioRxiv 2022, doi: <https://doi.org/10.1101/2022.06.01.493823>. Besides, the complete description of culturing conditions should be given per each sample. The media changed during culturing, but it is not clear if all samples within SFF and SFD conditions were cultured equally long in each medium or did the duration in different media varied across samples within each cultivation method e.g. based on the variation in time until confluence was reached in each sample. Similarly, were epithelial sheets exposed to same Ca concentrations for same time durations or did the Ca concentration varied between donors?

3. The authors conclude that both LAMB3+ basal keratinocytes and ZNF90+ fibroblasts were more involved in the interactions with melanocytes in the SFF versus SFD epithelial sheets. These conclusions are based on the receptor-ligand analyses of the scRNA-seq data and further experimental data would strengthen the transcriptomic data. Given that receptor-ligand interactions happen between interacting cells, did LAMB3+ basal keratinocytes and ZNF90+ fibroblasts co-localize in epithelial sheets and healthy skin with melanocytes? Did melanocytes in SFF compared to SFD sheets e.g. produce a different amount of melanin or showed different activity states? Did shallow transcriptomes of sequenced melanocytes differ between the two epithelial sheets? It would also be helpful to integrate published scRNA-seq data from vitiligo skin

and healthy skin with author's own scRNA-seq data to explore frequencies of LAMB3+ basal keratinocytes and ZNF90+ fibroblasts decrease in vitiligo lesions compared to healthy skin? Additionally, it would be of interest to see whether fibroblast and keratinocyte subsets previously linked to immune cell recruitment and suppressing re-pigmentation could be differentially enriched in differentially cultivated epithelial sheets (see <https://www.nature.com/articles/s41586-021-04221-8> and <https://www.biorxiv.org/content/10.1101/2021.12.03.470971v1.full>).

4. The authors state in the abstract that their findings support the LAMB3+ basal keratinocytes and ZNF90+ fibroblasts are key factors behind the re-pigmentation in SFF epithelial sheets. Further experimental data are needed to support this statement. The proportion of melanocytes in SFF and SFD sheets appeared much higher than in the normal skin (Fig3e), could SFD and SFF epithelial sheets already contain a sufficient number of melanocytes that could support the observed re-pigmentation at 1 year. Alternatively, could any other cell types or their subtypes be involved in differential melanocyte activities in SFF and SFD sheets including suppression of depigmentation (see also the comment above)?

5. SFF and SFD culture conditions differed significantly, not only because of the presence/absence of feeder cells but also because of rather different composition of culture media (e.g. growth factors etc). The observed differences in cell composition between SFD and SFF sheets might be in part also a consequence of different culturing conditions or e.g. altered survival of specific cell types in different media. Differential culture conditions could affect cellular transcriptomes when compared to healthy skin and these differences should be analysed in the manuscript. The heatmap in Suppl. Fig 1e shows differential transcriptional signatures within cell type clusters, could some of these differences possibly reflect cultured versus healthy skin conditions?

6. The clinical follow-up period after the surgery involved several time points (1, 3, 6, 12 month follow up points) at which grafted sites were evaluated for different parameters, e.g. Woods lamp assessment, confocal microscopy, re-pigmentation, complications (manuscript lines 379 -380, 390-391). Why did the authors report only the re-pigmentation outcomes at 1 year? What are the expected timeline and kinetics of the re-pigmentation process during the follow-up period? Could a differential cell composition of the SFF and SFD sheets at the time of transplantation influence the speed of re-pigmentation at treated vitiligo sites, did the attachments of the SFF and SFD sheets differ? Did the authors observe any differences in the kinetics of re-pigmentation across the follow-up visits when comparing the SFF and SFD grafted sites? This data is not only of interest given the authors hypothesis, (ZNF90+ fibroblast and LAMB3+ basal keratinocytes facilitating re-pigmentation in melanocyte poorer SFF sheets) but could also be clinically important since a faster re-pigmentation would diminish psychological stress linked to the disease. Did the re-pigmentation patterns differ between the two epithelial sheet cultivation, based on the re-pigmentation patterning criteria described in section 8.6?

7. It is not clear how the 3 patients were selected for scRNA-seq analysis among all the patients, how many other patients in the total cohort would be eligible for scRNA-seq analysis based on the criteria reported by the authors (89-96). The selection process should be fully described.

Minor comments

1. Histology images are shown only for SFF samples, SFD samples should also be shown (Fig 3C, Suppl Fig 3).

2. The authors focus on description of modules enriched in ZNF90+ fibroblast and LAMB3+ basal keratinocytes. What were pathways/modules enriched in other keratinocyte and fibroblast subtypes, which biological processes were implied in these pathways/modules?

4. How does dopa staining for determination of melanocyte-keratinocyte ratio reflect the ratio of these cells in scRNA-seq datasets?

5. Slightly different target cell encapsulation numbers are reported in text and methods, could the authors correct. The full name of 10x genomics reagent kit should be provided (e.g. version number missing)

6. Figure 2A: could authors specify the Chinese medicine drugs, was donor area measured in cm²?

7. Fig. 4D: the distribution of modules 4 and 15 should be shown across all cells as shown in Fig 4a.

8. Suppl. Fig 1D shows SkinsheetOld 1-3 versus Skinsheet, what does old denote here?

9. Isotype control staining and unstained sample should be included in Figs 3c and Suppl Fig 3c.

Reviewer #3 (Remarks to the Author):

The manuscript by Lian et al, entitled 'Single-cell sequencing reveals increased LAMB3+ basal keratinocytes and ZNF90+ fibroblasts in autologous cultured epithelium under serum- and feeder-free conditions' aims to address an understudied area in vitiligo research, namely, mechanisms behind treatment response determinants. Numerous surgical approaches exist for stable vitiligo patients and can be broadly divided into tissue grafting methods (punch grafts, suction blisters, hair follicle grafts) and cellular grafting methods (cultured epidermal cells, non-cultured cells). The study follows gene expression at the single cell level and compares cells isolated from 3 patients, which generated 6 sets of epidermal cells cultured in different conditions (serum-, feeder-free (SFF) vs. serum-, feeder-dependent (SFD)) for 18 days, which were then used for grafting. The beauty of the study is that it is patient-focused, using patient-derived cells that were subsequently grafted onto the same patients. The authors identified increased proportions of Lamb3+ basal keratinocytes and Znf90+ fibroblasts in SFF-conditioned cells and they analyzed these cells using computational approaches and make the following conclusions: 1) "LAMB3+ basal keratinocytes may enhance repigmentation by exerting an influence on the skin microenvironment within the whole epidermal melanin units..." 2) "FN1 is an essential molecular marker for ZNF90+ fibroblasts, which facilitates repigmentation when transplanting cultured epithelial sheets onto patients with vitiligo". However, both SFF and SFD cultured cells yielded similar repigmentation results as clearly demonstrated by the authors in figure 2 so the relative decrease in Lamb3+ keratinocytes and Znf90+ fibroblasts do not seem to contribute to whether a patient responds to treatment or not. While culturing cells under SFF conditions may help minimize infection risks and are therefore desirable, the current data does not support any functional role for Lamb3+ keratinocytes and Znf90+ fibroblasts despite characterization of ligand-receptor pathways. Thus, the conclusions in the manuscript is overstated based on the data presented currently as they are mainly correlative in nature and based on comparing cells that have been cultured for almost 3 weeks in drastically different conditions. Furthermore, patient variability can affect data consistency and this can be further exacerbated by the culturing of these cells. As such, more quality control and functional data is necessary to assign causative roles for 'Lamb3+ keratinocytes' and 'Znf90+ fibroblasts'. There are some smaller questions to be answered but most importantly, there is a need for additional functional data to demonstrate any role for these keratinocyte and fibroblast subpopulations.

Point-by-Point Response to Reviewers

Reviewer #1

The manuscript is a well-constructed study with valuable information that will add to the related literature.

Response: We thank the reviewer for giving us an opportunity to revise this manuscript. All the reviewer's comments have been replied point-by-point.

Comment 1: line 94: vitiligo vulgaris Not vulgaris vitiligo

Response: Thank you for your careful review and for pointing out this error. We have made the necessary correction in the manuscript in line 129.

Comment 2: line 110: The 9 samples; please identify their details

Response: We appreciate your suggestion to provide more details, which will enhance the clarity of our paper. We have added the relevant details in line 151 and Supplementary Fig. S2a: "A total of 9 samples were collected: healthy skin from 3 patients and autologous cultured SFD and SFF epithelial sheets (Supplementary Fig. S2a)."

Comment 3: line 118: makers? Do you mean markers

Response: Thank you for catching the spelling error. We have rectified this in the manuscript in line 178

Comment 4: line 238: please add details on your previous study

Response: Thank you very much for your suggestion, which adds to the comprehensibility of our study and underscores the continuity of our research. We have included the requested information in line 383: "Notably, our modified autologous cultured epithelial sheets without serum and feeder, which showed no differences in repigmentation compared to Rheinwald and Green's approach, according to the case-control study by the authors."

Comment 5: line 366: please mention how de-epithelization was performed

Response: Thank you for your expert suggestion; this detail is of significant interest to many dermatologists. We have provided the necessary information in line 570: "Before grafting, recipient areas were de-epithelialized through dermabrasion."

Comment 6: Please add Wood's light photos of recipient sites

Response: We acknowledge the value of Wood's lamp photos as you pointed out, but regrettably, we were unable to provide these images due to equipment limitations. In our study, the primary assessment was made through the visual examination by two dermatologists, with hand-held Wood's lamp and reflectance confocal microscopy serving as auxiliary diagnostic tools. While Wood's lamp photos were not taken due to equipment constraints, we do have reflectance confocal microscopy images, as shown in Supplementary Fig. S1. We have also clarified this aspect in the methods section in line 596: "Repigmentation rates were primarily assessed by two dermatologists through independent visual examination, with Wood's lamp and reflectance confocal microscopy serving as auxiliary diagnostic tools."

Comment 7: Please add a table summarizing patients' demographic and clinical data

Response: Thank you for your valuable suggestion. We have enhanced our manuscript by adding a table summarizing the patients' demographic and clinical data based on the information collected in the study questionnaire. The addition includes race, classification, duration of disease, comorbidities, family history, and post-operative repigmentation efficacy and adverse events, as seen in the table of Fig. 2a.

Comment 8: The paper needs professional English editing

Response: We appreciate your feedback regarding English editing. We have utilized the professional editing service provided by Springer Nature, and we have included a certificate of this service in the Supplementary File section. We hope that the edited version of the paper will make for a smoother and more convenient reading experience.

Reviewer #2

The manuscript by Jinhua Xu and coauthors contains valuable and unique human samples and datasets, with potential value for clinical translation and biological discovery. The paper contains novel results and could be of significant interest for scientific community beyond the field of vitiligo studies.

My major concern is that the currently presented experimental results do not provide enough supportive evidence for the conclusions of the manuscript since the data are mainly descriptive. Further experimental evidence and expansion of clinical data would be beneficial to further support the central hypothesis and conclusions. The manuscript would benefit from providing more methodological details to facilitate the reproduction of experiments by other scientists.

Response: We sincerely appreciate Reviewer #2's thorough and invaluable feedback. The comprehensive nature of the suggestions and the hands-on guidance provided by the reviewer have been instrumental in improving the quality of our manuscript. Below, we address each of the reviewer's comments in detail to incorporate these constructive recommendations.

Major concerns:

Comment 1: In the provided informed consent template form (supplementary data), patient consented to the collection, processing, analysing and publishing their anonymized nonidentifiable clinical and demographic data. Could the authors provide also the informed consent form templates for tissue analysis by scRNA-seq (where SNPs could be identifiable) and other experiments or describe in manuscript how these studies were regulated.

Response: Thank you for your reminder. We have provided two ethics approval reference numbers for our research. Ethics approval KY2020-698 was obtained for the retrospective study of autologous cultured epithelium grafting in the treatment of vitiligo, which involved recruiting vitiligo patients for clinical research on skin grafting. This cohort is still ongoing. Ethics approval KY2020-1137 was obtained for "The mechanism of the repigmentation of tissue engineering autologous cultured epithelium grafting in the treatment of vitiligo," which is a part of our research involving sequencing. We have included the relevant documents in the Supplementary File. S4, which include (a) the study protocol, (b) approval from the Ethics Review Committee of Huashan Hospital, Fudan University, and (c) informed consent forms for research and sample usage in sequencing and other experiments. We have also added the ethics approval reference numbers in the Methods section in line 529.

Comment 2: The graphic representation of scRNA-seq data (QC and results) should be improved.

(1) The authors should report cell number contribution of each experimental condition (SFF, SFD and healthy skin) to the total integrated dataset, please see as example also Alivernini S. et al, Nat Med 2020, Extended data Fig 2a).

Response: Thank you for your suggestion, and for providing the reference. We have now included the cell number contributions for each experimental condition (SFF, SFD, and healthy skin) in Supplementary Fig. S2a. As indicated in the results in line 151, “a total of 9 samples were collected, including healthy skin from 3 patients and autologous cultured SFD and SFF epithelial sheets (Supplementary Fig. S2a).”

(2) QC of scRNA-seq data should be shown per each sequenced sample (n=9), and the same is true for the results in Fig 3. In this way, the cell type variability across and within treatment groups will be fully apparent, please see as example also Figure 2b,c and Suppl Fig. 1a in Edalat S et al, bioRxiv 2022, doi: <https://doi.org/10.1101/2022.06.01.493823>.

Response: We appreciate your suggestion and the reference provided. We have created QC plots for each individual sequenced sample to demonstrate the quality control results, as shown in Supplementary Fig. S2d. Furthermore, we have incorporated the proportion of cell types for each sample in Fig 3d, as you recommended, which provides a more comprehensive view of our results.

(3) Besides, the complete description of culturing conditions should be given per each sample. The media changed during culturing, but it is not clear if all samples within SFF and SFD conditions were cultured equally long in each medium or did the duration in different media varied across samples within each cultivation method e.g. based on the variation in time until confluence was reached in each sample.

Response: Thank you for your valuable feedback regarding the culture conditions. The culturing times for each patient's epithelial sheets were indeed slightly different, based on the time it took for the sheets to reach confluence. In our study, the duration of culturing for three patients was quite consistent, as outlined in Fig. 2a. The duration until confluence for patient 1 was 8 days for both SFD and SFF, for patient 2 it was 7 days for both SFD and SFF, and for patient 3 it was 8 days for both SFD and SFF. Total culture days of SFF and SFD from three patients were all 18 days. This uniformity in the samples to a large extent reduces the heterogeneity introduced by the sequencing process.

(4) Similarly, were epithelial sheets exposed to same Ca concentrations for same time durations or did the Ca concentration varied between donors?

Response: We are grateful for your question about the calcium concentrations in the culturing conditions. In our study, the calcium concentrations were consistent across different donors. We have now added specific information about Ca concentrations and timing to the Methods section in line 559: “The Ca²⁺ concentration was 0.15 mM on Day 1, 0.5 mM on Days 2 to 3, 1.0 mM on Days 4 to 5, and 1.4 mM on Days 6 to 7, which was maintained until Day 10.”

Comment 3: The authors conclude that both LAMB3+ basal keratinocytes and ZNF90+ fibroblasts were more involved in the interactions with melanocytes in the SFF versus SFD epithelial sheets. These conclusions are based on the receptor-ligand analyses of the scRNA-seq data and further experimental data would strengthen the transcriptomic data.

(1) Given that receptor-ligand interactions happen between interacting cells, did LAMB3+ basal keratinocytes and ZNF90+ fibroblasts co-localize in epithelial sheets and healthy skin with melanocytes?

Response: Thank you for your valuable suggestion. To further support our conclusions, we have recruited two additional patients in an ongoing study. We have conducted immunofluorescence staining on their SFF and SFD epithelial sheets, co-localizing LAMB3+ basal keratinocytes and ZNF90+ fibroblasts with melanocytes in both the epithelial sheets and healthy skin, as described in Fig. 6e and line 361, “Immunofluorescence staining demonstrated that these two clusters co-localized with melanocytes in SFF, SFD epithelial sheets and healthy skin (Fig. 6e)”, which strengthens our findings.

(2) Did melanocytes in SFF compared to SFD sheets e.g. produce a different amount of melanin or showed different activity states? Did shallow transcriptomes of sequenced melanocytes differ between the two epithelial sheets?

Response: We appreciate your suggestion. We have analyzed melanocyte activity in SFF and SFD sheets using GSVA on the scales of cells and found that melanocytes in SFF exhibited higher activity in melanin production and melanin differentiation compared to SFD, as shown in Fig. 3g. However, there were no significant differences in melanocyte proliferation. While GSEA on the scales of groups did not reveal any significant differences in genes related to melanocytes between the two types of epithelial sheets. This information has been added to the Supplementary Materials and mentioned in the results at line 220. : “In regard to melanocytes, which can directly affect repigmentation after grafting, GSVA (Gene set variation analysis) using Reactome and GO database showed that pathways including “melanin biosynthesis”, “PI3K AKT signaling”, “cell-cell adhesion”, and “positive regulation of melanocyte differentiation” were enhanced in melanocytes in SFF epithelial sheets compared to SFD epithelial sheets; there was no difference in “melanocyte proliferation” (Fig. 3g; Supplementary File. S1). On the scale of samples, GSEA (Gene set enrichment analysis) of sequenced melanocytes showed no difference between the two epithelial sheets, enrichment analysis also showed no difference in melanocyte-related pathways (Supplementary Fig. S5; Supplementary File. S1). Thus, melanocytes in SFF sheets may exhibit a higher activity in melanin production and differentiation compared to those in SFD sheets, which could potentially be attributed to the variations observed in cell populations.”

(3) It would also be helpful to integrate published scRNA-seq data from vitiligo skin and healthy skin with author’s own scRNA-seq data to explore frequencies of LAMB3+ basal keratinocytes and ZNF90+ fibroblasts decrease in vitiligo lesions compared to healthy skin?

Response: This idea is intriguing. We have integrated scRNA-seq data from Xu et al. and calculated the frequencies of LAMB3+ basal keratinocytes and ZNF90+ fibroblasts in vitiligo lesions compared to healthy skin. We observed a decrease in the proportion of LAMB3+ keratinocytes in vitiligo lesions, This decrease may be indicative of their correlation with skin pigmentation, as mentioned in the results at line 193: “Then, we explored the frequencies of these two clusters in scRNA-seq data from vitiligo lesions and healthy skin, and found a decrease in the proportion of LAMB3+ basal keratinocytes in lesions (lesions vs. normal skin: 3.19% vs. 4.32%, $P>0.05$) and no difference in the proportion of ZNF90+ fibroblasts (lesions vs. normal skin: 1.12% vs. 0.63%, $P<0.05$) (Supplementary Fig. S4a), which may reflect the correlation between these two clusters of cells and skin pigmentation”.

(4) Additionally, it would be of interest to see whether fibroblast and keratinocyte subsets previously linked to immune cell recruitment and suppressing re-pigmentation could be differentially enriched in differentially cultivated epithelial sheets (see <https://www.nature.com/articles/s41586-021-04221-8> and <https://www.biorxiv.org/content/10.1101/2021.12.03.470971v1.full>).

Response: These are also very helpful suggestions. We have examined the enrichment of fibroblast and keratinocyte subsets associated with immune cell recruitment and suppressing re-pigmentation, as suggested by the references provided. We have added this information to the results at line 198: “Prior research has proposed that KRT6A+ “stress keratinocytes” and Ifngr1+ fibroblasts are implicated in the immune response associated with vitiligo. However, notably, our dataset did not reveal enrichment of Ifngr1+ fibroblasts in the epithelial sheets (Supplementary Fig. S4b), compared to healthy skin; stress keratinocytes constituted a small proportion in the epithelial sheets (Supplementary Fig. S4c). This observation contributes to the establishment of a foundational immune-related rationale for repigmentation.”

Comment 4: The authors state in the abstract that their findings support the LAMB3+ basal keratinocytes and ZNF90+ fibroblasts are key factors behind the re-pigmentation in SFF epithelial sheets. Further experimental data are needed to support this statement. The proportion of melanocytes in SFF and SFD sheets appeared much higher than in the normal skin (Fig3e), could SFD and SFF epithelial sheets already contain a sufficient number of melanocytes that could support the observed re-pigmentation at 1 year. Alternatively, could any other cell types or their subtypes be involved in differential melanocyte activities in SFF and SFD sheets including suppression of depigmentation (see also the comment above)?

Response: Thank you for your suggestion. Based on our existing experimental results, we agree that describing LAMB3+ basal keratinocytes and ZNF90+ fibroblasts as "key factors" in the abstract might have been an exaggeration. Therefore, we have amended the abstract to refer to them as "positive factors." Additionally, we conducted supplementary experiments by recruiting 2 patients who consented to use their biopsy for experimental use. Flow cytometry revealed that the proportion of LAMB3+ keratinocytes and ZNF90+ fibroblasts in our epithelial sheets is relatively low, making it impractical to sort these two cell populations using flow cytometry. Consequently, we directly dissociated the normal skin and cultured keratinocytes in keratinocyte medium and fibroblasts in fibroblast medium. We then knocked down LAMB3 in keratinocytes and ZNF90 in fibroblasts using lentivirus transfection. Subsequently, we conducted qPCR analysis of genes related to melanocyte receptor-ligand interactions. The results indicate that the knockdown cells exhibited downregulation in the expression of melanocyte receptor-ligand-related genes. This provides some degree of validation for the importance of these two cell populations in melanocyte-related pathways.

In another unpublished article by us, we compared single-cell sequencing data from patients with vitiligo skin lesions, normal skin, epithelial sheets, and post-repigmentation skin. We demonstrated that, in some patients, a higher number of melanocytes in the sheets did not necessarily correlate with better repigmentation outcomes. Conversely, some patients with fewer melanocytes achieved better repigmentation. This suggests that factors beyond melanocytes may play a significant role in repigmentation. Existing literature highlights the supportive role of basal keratinocytes and fibroblasts in melanocyte function. In alignment with our single-cell sequencing results, we observed that the population of LAMB3+ basal keratinocytes and ZNF90+ fibroblasts in the sheets was relatively higher. Although other cell types may also support melanocytes, our study did not reveal significant differences in their contribution within the sheets. Further experiments may be required to investigate other cell types' roles within the sheets.

Comment 5: SFF and SFD culture conditions differed significantly, not only because of the presence/absence of feeder cells but also because of rather different composition of culture media (e.g.

growth factors etc). The observed differences in cell composition between SFD and SFF sheets might be in part also a consequence of different culturing conditions or e.g. altered survival of specific cell types in different media. Differential culture conditions could affect cellular transcriptomes when compared to healthy skin and these differences should be analysed in the manuscript. The heatmap in Suppl. Fig 1e shows differential transcriptional signatures within cell type clusters, could some of these differences possibly reflect cultured versus healthy skin conditions?

Response: Thank you for your valuable suggestion. We have discussed, in the Discussion section, the impact of different culture conditions on cellular transcriptomes compared to healthy skin. We have also explored how certain cell compositions may change due to the culturing conditions, in line 394: "Considering that cellular transcriptomes can be affected under different culture conditions with different compositions of medium, we compared the cell compositions of cultured epithelium sheets with those of healthy skin. One of the findings was a decrease in the proportion of immune cells in the SFD and SFF sheets, which may be due to the absence of stimulators in the culture medium to maintain the growth of immune cells. However, this low immunogenicity favors transplantation."

In the quality control section, we used the Harmony algorithm to calibrate gene expression. The UMAP plots before and after calibration are presented in Supplementary Fig. S2e. These plots demonstrate that after calibration, the differences were reduced. The heatmap in suppl fig 1e served as one of the bases for single-cell sequencing clustering, rather than difference among groups. The difference caused by culture condition has been presented as cell proportion in Fig.3d, e.

Comment 6: The clinical follow-up period after the surgery involved several time points (1, 3, 6, 12 month follow up points) at which grafted sites were evaluated for different parameters, e.g. Woods lamp assessment, confocal microscopy, re-pigmentation, complications (manuscript lines 379 -380, 390-391).

(1) Why did the authors report only the re-pigmentation outcomes at 1 year?

Response: We would like to sincerely apologize for the limitation in our study design. Our original plan included follow-up at these time points, but due to factors such as long distances, strict pandemic control measures, and other logistical challenges, patients were lost to follow-up. As a result, we can only provide baseline and 12-month data. We have now updated our methods in line 139: "In this study, both methods achieved excellent (full repigmentation) results at the 1-year follow-up after transplantation (Fig. 2d; Supplementary Fig. S1). The kinetics of repigmentation were unable to examine some patients' loss to follow-up at 1, 3, and 6 months due to unforeseen circumstances."

(2) What are the expected timeline and kinetics of the re-pigmentation process during the follow-up period?

Response: Thank you for this excellent question. This is indeed of great interest to dermatologists and vitiligo researchers. We have included the expected timeline and kinetics in line 137: "Based on the repigmentation observed in previous patients, a one-month recovery period following grafting was followed by visible repigmentation at approximately 3 months, with the optimal outcome typically achieved within one year."

(3) Could a differential cell composition of the SFF and SFD sheets at the time of transplantation influence the speed of re-pigmentation at treated vitiligo sites, did the attachments of the SFF and SFD sheets differ? Did the authors observe any differences in the kinetics of re-pigmentation across the follow-up visits when comparing the SFF and SFD grafted sites?

Response: We believe that the cell composition and attachment of the graft sheets should be critical factors influencing re-pigmentation speed. However, based on our one-year follow-up results and clinical observations, we have not identified significant differences in the rate of repigmentation or graft attachment between the SFF and SFD sheets. Unfortunately, due to patients being lost to follow-up at various time points, we do not have definitive kinetic data to support our clinical observations. We have addressed this in line 141: “The kinetics of repigmentation were unable to examine some patients’ loss to follow-up at 1, 3, and 6 months due to unforeseen circumstances.”

(4) This data is not only of interest given the authors hypothesis, (ZNF90+ fibroblast and LAMB3+ basal keratinocytes facilitating re-pigmentation in melanocyte poorer SFF sheets) but could also be clinically important since a faster re-pigmentation would diminish psychological stress linked to the disease. Did the re-pigmentation patterns differ between the two epithelial sheet cultivation, based on the re-pigmentation patterning criteria described in section 8.6?

Response: Thank you for your meticulous review and raising this valuable clinical consideration. We agree that the speed and pattern of re-pigmentation are clinically significant and can significantly impact the psychological well-being of vitiligo patients. We have addressed this in line 142: "The repigmentation patterns are characterized by uniform diffuse repigmentation, with no significant differences observed between the 2 sheets. This uniform pattern is attributed to ACEG, which replaces the epidermal layer without modifying or stimulating the hair follicles."

Comment 7: It is not clear how the 3 patients were selected for scRNA-seq analysis among all the patients, how many other patients in the total cohort would be eligible for scRNA-seq analysis based on the criteria reported by the authors (89-96). The selection process should be fully described.

Response: Thank you for your comment. We have clarified the selection process for these three patients in our results in line 106: "From November 2020 to January 2021, we approached 17 eligible patients for inclusion in our study cohort, and only three individuals consented to contribute their samples for the investigation."

Minor comments:

Comment 1: Histology images are shown only for SFF samples, SFD samples should also be shown (Fig 3C, Suppl Fig 3).

Response: Thank you for pointing this out. We have included histology images for SFD samples in Fig 3c and Supplementary Fig S3.

Comment 2: The authors focus on description of modules enriched in ZNF90+ fibroblast and LAMB3+ basal keratinocytes. What were pathways/modules enriched in other keratinocyte and fibroblast subtypes, which biological processes were implied in these pathways/modules?

Response: We appreciate your suggestion. We have now conducted the GSVA analysis in other keratinocyte and fibroblast subtypes and included this information in the supplementary Fig. S6 and Supplementary File. S2.

Comment 3:

No third comment.

Comment 4: How does DOPA staining for determination of melanocyte-keratinocyte ratio reflect the ratio of these cells in scRNA-seq datasets?

Response: The DOPA staining is used to determine the melanocyte-keratinocyte (KM) ratio, serving as a quality control and standard for our cultured skin sheets. The relatively close KM ratios of the three patients helped ensure consistency in our subsequent sequencing experiments.

Comment 5: Slightly different target cell encapsulation numbers are reported in text and methods, could the authors correct. The full name of 10x Genomics reagent kit should be provided (e.g. version number missing).

Response: We have added the cell numbers of each sample in Supplementary Fig. S1a, that can match the information between the text and methods regarding target cell encapsulation numbers. We've also provided the full name of the 10x Genomics reagent kit including its version number in line 605.

Comment 6: Figure 2A: could authors specify the Chinese medicine drugs, was donor area measured in cm²?

Response: Thank you for your attention to Chinese medicine. We have specified the Chinese medicine drugs in the Fig.2a, that is “Traditional Chinese Medicine, a hospital-made herbal formulation named “Qibai Granules”, which includes ingredients such as Astragalus membranaceus (Huangqi) and Angelica dahurica (Baizhi).” And yes, the donor area was measured in cm².

Comment 7: Fig. 4D: the distribution of modules 4 and 15 should be shown across all cells as shown in Fig 4a.

Response: Thank you for highlighting this. We've added the distribution of modules 4 and 15 across all cells, as suggested.

Comment 8: Suppl. Fig 1D shows SkinsheetOld 1-3 versus Skinsheet, what does 'Old' denote here?

Response: Thank you for pointing this out. 'Old' denotes SFD skin sheets, and we have revised the denotation to 'SFF' and 'SFD' for clarity.

Comment 9: Isotype control staining and unstained sample should be included in Figs 3c and Suppl Fig 3c.

Response: Thank you for pointing this out. We have included isotype control staining and an unstained sample in Fig. 3c and Supplementary Fig S3.

Reviewer #3

Comment: The manuscript by Lian et al, entitled “Single-cell sequencing reveals increased LAMB3⁺ basal keratinocytes and ZNF90⁺ fibroblasts in autologous cultured epithelium under serum- and feeder-free conditions” aims to address an understudied area in vitiligo research, namely, mechanisms behind treatment response determinants. Numerous surgical approaches exist for stable vitiligo patients and can be broadly divided into tissue grafting methods (punch grafts, suction blisters, hair follicle grafts) and cellular grafting methods (cultured epidermal cells, non-cultured cells). The study follows gene expression at the single cell level and compares cells isolated from 3

patients, which generated 6 sets of epidermal cells cultured in different conditions (serum-, feeder-free (SFF) vs. serum-, feeder-dependent (SFD)) for 18 days, which were then used for grafting. The beauty of the study is that it is patient-focused, using patient-derived cells that were subsequently grafted onto the same patients. The authors identified increased proportions of Lamb3+ basal keratinocytes and Znf90+ fibroblasts in SFF-conditioned cells and they analyzed these cells using computational approaches and make the following conclusions: 1) “LAMB3+ basal keratinocytes may enhance repigmentation by exerting an influence on the skin microenvironment within the whole epidermal melanin units...” 2) “FN1 is an essential molecular marker for ZNF90+ fibroblasts, which facilitates repigmentation when transplanting cultured epithelial sheets onto patients with vitiligo”. However, both SFF and SFD cultured cells yielded similar repigmentation results as clearly demonstrated by the authors in figure 2 so the relative decrease in Lamb3+ keratinocytes and Znf90+ fibroblasts do not seem to contribute to whether a patient responds to treatment or not. While culturing cells under SFF conditions may help minimize infection risks and are therefore desirable, the current data does not support any functional role for Lamb3+ keratinocytes and Znf90+ fibroblasts despite characterization of ligand-receptor pathways. Thus, the conclusions in the manuscript is overstated based on the data presented currently as they are mainly correlative in nature and based on comparing cells that have been cultured for almost 3 weeks in drastically different conditions. Furthermore, patient variability can affect data consistency and this can be further exacerbated by the culturing of these cells. As such, more quality control and functional data is necessary to assign causative roles for ‘Lamb3+ keratinocytes’ and ‘Znf90+ fibroblasts’. There are some smaller questions to be answered but most importantly, there is a need for additional functional data to demonstrate any role for these keratinocyte and fibroblast subpopulations.

Response: Thank you for your appreciation of our study and for taking the time to provide feedback. We became interested in exploring the mechanisms behind ACEG repigmentation patterns through clinical observations, which led us to investigate the single-cell transcriptomes of skin sheets produced by two different methods. While our findings indicated that SFF-conditioned cells had fewer melanocytes compared to SFD-conditioned cells, we observed no significant differences in repigmentation outcomes. Therefore, our initial hypothesis was based on the reduced melanocytes in SFF, rather than a decrease in LAMB3+ keratinocytes and ZNF90+ fibroblasts in SFD. We appreciate your insightful comments and have addressed these concerns. To clarify:

(1) Quality Control: We have improved quality control data, displayed by sample, in Fig. 3d and Supplementary Fig. S2b, c, d, e. Despite variations due to different culturing conditions, our quality control measures show that we used a dataset with consistently high data quality for our analyses.

(2) Verification of Cell Types: To address concerns regarding cell type heterogeneity, we recruited two additional patients who provided consent, and we cultured SFF and SFD epithelium for experimental purposes. Through flow cytometry, we confirmed the presence of both these cell populations in their epithelial sheets, as shown in Fig. 3f. Immunofluorescence staining confirmed their co-localization with melanocytes, as shown in Fig. 6d. Thus, we were able to consistently culture the epithelium with these two cell populations, validating the quality of our experimental data. This information is now included in our manuscript at line 191: “By flow cytometry, we verified that the proportions of LAMB3+ basal keratinocyte clusters and ZNF90+ fibroblast clusters increased in SFF epithelial sheets compared to SFD

epithelial sheets (Fig. 3f)”, and line 361: “Immunofluorescence staining demonstrated that these two clusters co-localized with melanocytes in SFF, SFD epithelial sheets and healthy skin (Fig. 6e).”

(3) Functional Experiments: Due to the relatively low proportion of LAMB3⁺ keratinocytes and ZNF90⁺ fibroblasts detected by flow cytometry in our skin sheets, we were unable to directly sort these cell populations. Therefore, we opted to directly isolate the skin and culture them separately in keratinocyte and fibroblast medium. We knocked down LAMB3 in keratinocytes and ZNF90 in fibroblasts using lentiviral transfection. Although we attempted to co-culture these knockout cells with melanocytes, the strong proliferative capacity of melanocytes made it challenging to observe significant differences in melanin production. However, we conducted qPCR analysis for relevant genes in shLAMB3 keratinocytes and shZNF90 fibroblasts in relation to melanocyte receptor-ligand interactions, and the results indicated a downregulation of melanocyte receptor-ligand-related genes in these knockout cells. This partially validated our findings, suggesting the positive roles of these two cell populations in melanocyte-related pathways.

We appreciate your thoughtful review, and we have made every effort to address your concerns and improve the quality of our manuscript. Thank you for your guidance throughout the review process.

Update of figures:

1. Fig.2a: addition includes race, classification, duration of disease, comorbidities, family history, and post-operative repigmentation efficacy and adverse events, and the ingredients of mentioned Traditional Chinese Medicine for patients with vitiligo.
2. Fig.3c: added the staining panel of SFD epithelial sheets.
3. Fig.3d: added the presentation of the proportion of cell types by each sample.
4. Fig.3f: added the flow cytometry results of the proportion of LAMB3⁺ cells and ZNF90⁺ cells in SFF and SFD epithelial sheets.
5. Fig. 3g: added the GSVA results of melanocytes compared SFF to SFD epithelial sheets.
6. Fig.4d: added the distribution of modules 4 and 15 across all cells.
7. Fig. 6d: added the immunofluorescence staining of co-localization of LAMB3⁺ keratinocytes and ZNF90⁺ fibroblasts with melanocytes in SFF, SFD epithelium and normal skin.
8. Fig.6e: added the qPCR results showing expression levels of melanocyte-related ligand-receptor genes in keratinocytes with LAMB3 knockdown and fibroblasts with ZNF90 knockdown.
9. Supplementary Fig. S1: added the reflectance confocal microscopy images of three patients.
10. Supplementary Fig. S2a: added the cell distribution of each sample.
11. Supplementary Fig. S2d: added the quality control of each sample.
12. Supplementary Fig. S2e: added the UMAP before and after the quality control.
13. Supplementary Fig. S3: added the isotype and unstained control of samples.
14. Supplementary Fig. S4: added the integrative analysis with published dataset of vitiligo.
15. Supplementary Fig. S5: added shallow transcriptomes of sequenced melanocytes between two sheets.
16. Supplementary Fig. S6: added GSVA of other keratinocytes and fibroblasts.
17. Supplementary Fig. S7 added the results of knockdown of LAMB3 in keratinocytes and ZNF90 in fibroblasts.

REVIEWERS' COMMENTS:

Reviewer #1 (Remarks to the Author):

Dear authors,
Thank you for addressing the suggested concerns
The paper is now excellent
Best regards

Reviewer #2 (Remarks to the Author):

The study represents a beautiful example of bedside to bench to bedside approach. The authors put significant efforts to address my concerns and improved the data interpretation. Representation of scRNA-seq data and QC of scRNA-seq analysis is clearer. Knockout studies are highly appreciated, their results however, do not stronger support the roles of LAMB3+ keratinocytes and ZNF90+ fibroblasts in facilitating repigmentation in SFF sheets..

The principal message of the study is that culture conditions may affect strongly the properties of the cell type/subtype composition in the sheets, however these differences do not appear to affect the clinical efficacy of repigmentation. Thus, SFF sheets seem to be similarly efficient compared to SFD sheets, while they might be potentially safer.

Whether the differential cellular composition of the two sheets contributes to different mechanisms of repigmentation remains fully speculative. There could be similar or different factors in SFD and SFF sheets driving repigmentation. E.g. there is also an increase in granular keratinocytes in SFD sheets, could they have also a role in driving success of repigmentation in SFD grafts?

I would suggest that the authors recenter the interpretation of their main clinical findings, while utilising scRNA-seq data for hypothesis generation.

Minor:

1. I suggest a second round of English proofreading, and a detailed check of the Figures for errors e.g. NUE in Fig 3b - did the authors mean NEU as abbreviation for neuron.
2. How do authors explain a large difference in % of LAMB3+ cells between flow cytometry and scRNA-seq?
3. Sress 2 keratinocytes seem to present a large and not small proportion of keratinocytes in SFD sheets? Is the scale correct?

Reviewer #3 (Remarks to the Author):

The manuscript by Lian et al, entitled 'Single-cell sequencing reveals increased LAMB3+ basal keratinocytes and ZNF90+ fibroblasts in autologous cultured epithelium under serum- and feeder-free conditions' aims to address an understudied area in vitiligo research, namely, mechanisms behind treatment response determinants. By improving figures, quality control and including additional data, the authors have adequately addressed most previous concerns and questions.

Some additional comments:

- Line 141-143: "The kinetics of repigmentation..." sentence needs to be restructured.
- Line 219-220 "This observation contributes to the establishment of a foundational immune-related rationale for repigmentation." Unclear what observation the authors are referring to and what they mean by foundational immune-related rationale.
- The interactions with melanocytes between Lamb3+ keratinocytes and Znf90+ fibroblasts were interesting especially with the knockdown experiments. Given the lack of these populations in SFD conditions, why do SFD cultured-cells provide similar repigmentation results? Do they speculate this will change with time (i.e durability of response maybe different?)?

Point-by-Point Response to Reviewers

Reviewer #1 (Remarks to the Author):

Dear authors,

Thank you for addressing the suggested concerns

The paper is now excellent

Best regards

Response: Thank you for your suggestions and feedback, and for your time.

Reviewer #2 (Remarks to the Author):

The study represents a beautiful example of bedside to bench to bedside approach. The authors put significant efforts to address my concerns and improved the data interpretation. Representation of scRNA-seq data and QC of scRNA-seq analysis is clearer. Knockout studies are highly appreciated, their results however, do not stronger support the roles of LAMB3+ keratinocytes and ZNF90+ fibroblasts in facilitating repigmentation in SFF sheets..

The principal message of the study is that culture conditions may affect strongly the properties of the cell type/subtype composition in the sheets, however these differences do not appear to affect the clinical efficacy of repigmentation. Thus, SFF sheets seem to be similarly efficient compared to SFD sheets, while they might be potentially safer.

Whether the differential cellular composition of the two sheets contributes to different mechanisms of repigmentation remains fully speculative. There could be similar or different factors in SFD and SFF sheets driving repigmentation. E.g. there is also an increase in granular keratinocytes in SFD sheets, could they have also a role in driving success of repigmentation in SFD grafts?

I would suggest that the authors recenter the interpretation of their main clinical findings, while utilising scRNA-seq data for hypothesis generation.

Response: Thank you for your invaluable suggestions throughout, and for your concise summary of the paper. We have made revisions in response to your recommendations. In the abstract (lines 46-48): " Overall, SFF sheets demonstrate comparable efficacy to SFD sheets, offering superior safety. LAMB3+ basal keratinocytes and ZNF90+ fibroblasts act as potential drivers behind repigmentation in ACEG under SFF conditions." Additionally, in the discussion section, we addressed differences in cellular composition due to varied culture conditions, introducing it at the outset (line 289): "The serum and feeder of the SFD-conditioned sheets can support the formation of the graft, while the SFF sheets exhibit potential advantages in safety." Further, in line 302, we postulate based on single-cell analysis: "and the subsequent analysis may support the hypothesis that these 2 clusters facilitate the function of melanocytes on the basis of the lower cell proportion of melanocytes in SFF sheets compared to SFD sheets." To summarize our discussion (lines 368-371): "Culture conditions, while strongly influencing the properties of the cell composition in the sheets, do not appear to affect the clinical efficacy of repigmentation. This suggests that SFF sheets exhibit similar efficiency to SFD sheets while offering enhanced safety, marking a promising avenue for further clinical application."

Regarding your mention of other factors driving repigmentation, we acknowledge their presence, yet our focus remains on the two cell populations predominant in SFF over SFD. We previously discussed granular keratinocytes' role in increasing graft thickness but omitted it for coherency. In essence, we believe granular keratinocytes contribute to post-transplant skin reconstruction due to their involvement in FGFR, BMP, and WNT signaling, closely linked to embryonic epidermal morphogenesis and melanocyte activation, suggesting their partial role in driving repigmentation.

Minor:

Comment 1: I suggest a second round of English proofreading, and a detailed check of the Figures for errors e.g. NUE in Fig 3b - did the authors mean NEU as abbreviation for neuron.

Response: Thank you for your advice. After another round of proofreading, we've identified and addressed the similar issues.

Comment 2: How do authors explain a large difference in % of LAMB3+ cells between flow cytometry and scRNA-seq?

Response: Thank you for pointing this out. We have carefully considered the issue and would like to address it from several perspectives. Firstly, we have tried our best to replicate the experiments with limited samples, regrettably, and it turned out to be similar results. We acknowledge the presence of technical limitations, particularly concerning LAMB3 expression primarily within the cytoplasm, and we suppose that some loss of cell may be caused due to fixation and membrane permeabilization during flow cytometry, so cytoplasmic markers might be less efficiently detected. Secondly, single-cell RNA sequencing can more comprehensively detect cell types based on gene expression, including those with cytoplasmic markers, which might not be as efficiently detected by flow cytometry. Lastly, our use of flow cytometry successfully identified the presence and differences in expression of LAMB3+ keratinocytes and ZNF90+ fibroblasts between SFF and SFD sheets, aligning with our intended methodology.

Comment 3: Stress 2 keratinocytes seem to present a large and not small proportion of keratinocytes in SFD sheets? Is the scale correct?

Response: Thank you for your thorough review. We have checked the scale and found it correct. We suppose that, during in vitro culture, cells may undergo stress, resulting in the prevalent expression of KRT6A. However, both stress keratinocyte populations are lower in SFF sheets. We have revised the manuscript accordingly, stating: "stress keratinocytes constituted a small proportion in the SFF epithelial sheets, while a certain proportion of stress 2 subpopulation was observed in the SFD epithelial sheets (Supplementary Fig. S5c). This suggests that the proportion of aberrant cell subpopulations associated with immune response in vitiligo lesions was minor in our SFF epithelial sheets." (line 153-156).

Reviewer #3 (Remarks to the Author):

The manuscript by Lian et al, entitled 'Single-cell sequencing reveals increased LAMB3+ basal keratinocytes and ZNF90+ fibroblasts in autologous cultured epithelium under serum- and feeder-

free conditions” aims to address an understudied area in vitiligo research, namely, mechanisms behind treatment response determinants. By improving figures, quality control and including additional data, the authors have adequately addressed most previous concerns and questions.

Thank you for your time and feedback. Here are our responses to the comments:

Some additional comments:

Comment: Line 141-143: "The kinetics of repigmentation..." sentence needs to be restructured.

Response: We appreciate your suggestion. We've restructured the sentence in lines 108-110: "The kinetics of repigmentation were unable to examine as a result of the loss of follow-up for certain patients at 1, 3, and 6 months due to unforeseen circumstances. "

Comment: Line 219-220 "This observation contributes to the establishment of a foundational immune-related rationale for repigmentation." Unclear what observation the authors are referring to and what they mean by foundational immune-related rationale.

Response: Thank you for pointing this out. To enhance the scope of our study, particularly regarding the previously unaddressed immune response, we supplemented our data by integrating single-cell datasets from other vitiligo studies. This helped correlate cell populations associated with vitiligo immune responses to those present in our sheets, revealing a relatively low cell count. We acknowledge our previous statement's generality and have revised lines 154-156: "This suggests that the proportion of aberrant cell subpopulations associated with the immune response in vitiligo lesions was minor in our SFF epithelial sheets."

Comment: The interactions with melanocytes between Lamb3+ keratinocytes and Znf90+ fibroblasts were interesting especially with the knockdown experiments. Given the lack of these populations in SFD conditions, why do SFD cultured-cells provide similar repigmentation results? Do they speculate this will change with time (i.e durability of response maybe different?)?

Response: Thank you for your feedback. Our primary findings indicate that although different culture conditions alter cellular compositions, both SFF and SFD exhibit comparable clinical repigmentation effects. Interestingly, melanocyte decreased in SFF sheets, leading us to speculate on the correlation between LAMB3+ keratinocytes and ZNF90+ fibroblasts with effective

repigmentation despite reduced cell thickness and melanocyte presence in SFF sheets. Furthermore, SFF sheets demonstrated enhanced safety, suggesting potential for wider clinical application. Notably, SFD sheets didn't lack these cell populations but showed proportionally fewer than SFF. Repigmentation likely involves various factors, such as granular keratinocytes (higher proportion in SFD than SFF), yet upon comparing these two sheets, we propose these two cell populations may contribute to repigmentation without serum or feeder layers. According to our clinical observations, the repigmentation is stable with time.